# Green Design of Novel Starch-Based Packaging Materials Sustaining Human and Environmental Health

**DOI:** 10.3390/polym13081190

**Published:** 2021-04-07

**Authors:** Monica Mironescu, Andrada Lazea-Stoyanova, Marcela Elisabeta Barbinta-Patrascu, Lidia-Ioana Virchea, Diana Rexhepi, Endre Mathe, Cecilia Georgescu

**Affiliations:** 1Faculty of Agricultural Sciences Food Industry and Environmental Protection, Lucian Blaga University of Sibiu, 7-9 Ioan Ratiu Street, 550012 Sibiu, Romania; cecilia.georgescu@ulbsibiu.ro; 2National Institute for Lasers, Plasma and Radiation Physics, 409 Atomistilor Street, Magurele, 077125 Ilfov, Romania; 3Department of Electricity, Faculty of Physics, Solid-State Physics and Biophysics, University of Bucharest, 405 Atomistilor Street, P.O. Box MG-11, 077125 Bucharest-Magurele, Romania; 4Faculty of Medicine, Lucian Blaga University of Sibiu, 2A Lucian Blaga Street, 550169 Sibiu, Romania; lidia_virchea@yahoo.com; 5Faculty of Agricultural and Food Sciences and Environmental Management, University of Debrecen, H-4032 Debrecen, Hungary; dianarexhepi7@gmail.com (D.R.); endre.mathe64@gmail.com (E.M.); 6Faculty of Medicine, “Vasile Goldis” Western University of Arad, 310045 Arad, Romania

**Keywords:** green packaging, green chemistry, green technology, starch, bioactive, health, UV, cold plasma, nanopackaging, bioinspiration

## Abstract

A critical overview of current approaches to the development of starch-containing packaging, integrating the principles of green chemistry (GC), green technology (GT) and green nanotechnology (GN) with those of green packaging (GP) to produce materials important for both us and the planet is given. First, as a relationship between GP and GC, the benefits of natural bioactive compounds are analyzed and the state-of-the-art is updated in terms of the starch packaging incorporating green chemicals that normally help us to maintain health, are environmentally friendly and are obtained via GC. Newer approaches are identified, such as the incorporation of vitamins or minerals into films and coatings. Second, the relationship between GP and GT is assessed by analyzing the influence on starch films of green physical treatments such as UV, electron beam or gamma irradiation, and plasma; emerging research areas are proposed, such as the use of cold atmospheric plasma for the production of films. Thirdly, the approaches on how GN can be used successfully to improve the mechanical properties and bioactivity of packaging are summarized; current trends are identified, such as a green synthesis of bionanocomposites containing phytosynthesized metal nanoparticles. Last but not least, bioinspiration ideas for the design of the future green packaging containing starch are presented.

## 1. The “Green” Context

Politicians, organizations, companies and consumers around the world are increasingly aware that traditional food packaging, especially plastic goods, causes tremendous damage to the environment, water supplies and the entire ecosystem [1,2,3,4]. The food packaging field is still dominated by petroleum-derived polymers, such as polyethylene and polystyrene, despite global concerns about the environment [5]. As a result, in addition to other food packaging strategies such as the reduction, recycling and reuse of packaging, alternative “green” materials from renewable resources are now required [6,7]. 

The term “green” was developed in the 1980s as a business concept to use environmental issues as competitive advantages [8] and evolved into the modern and fashionable “green paradigm” [9]. Green packaging (GP) has its place inside this paradigm, as Figure 1 shows, and is connected with many other modern concepts. Zhang and Zhao [10] defined the green package as an “environmental friendly package, which is completely made by natural plants, can be circle or second use, be prone to degradation and promote sustainable development, even during its whole lifecycle, it is hurtless to environment as well as to human body and livestock’s health”. The strategies used in developing green packages comply with the twelve principles of Green Chemistry (GC) and with the Green Technology (GT) (both detailed in chapters 4 and 5 of this review). GP refers to a safe packaging design with low environmental impact; it is also known as environmentally friendly packaging or sustainable packaging [11]; new terms as ecobenign, biofriendly, biobenign are used [3]. 

Green economic examples in food packaging are: limiting the waste creation; maximizing sustainable materials (such as renewable, recyclable or biodegradable packages); use of renewable energy for packaging production [12]. The big producers in today’s green market are largely focusing on developing packaging that can be recycled (according to the strategy of circular economy, a practical green economic solution) [13,14,15] or biobased plastics (biodegradable or not) [16]. 

At the marketing level, GP can be obtained by using adequate logistic techniques (transport, storage, packaging, handling, processing and distribution [10]), adequate labels (i.e., “symbols printed on products or their packaging to advertise environmental quality or characteristics” [17]), green packaging marketing mix (price, product, promotion and place) and green strategies [18] (Figure 1).

GP can be implemented by applying green management and green politics, directly and indirectly promoted by governments, institutions or companies through adequate laws, regulations, or taxation, and by institutional rules and strategies at company level [10]. The economic aspects should be considered by designing sustainable packaging; consumers need to pay more for green packaging [19], and companies need to overcome their attitude to economic wellbeing and to start examining the GP actions, to change their attitudes and to establish future “green” plans [20]. As Figure 1 shows, education is directly linked to the economy, being necessary to develop positive consumer beliefs and behaviors related with packaging; recent studies showed that consumers are ready to pay more for sustainable GP [11,19]. 

Traditionally, food-packaging materials have been chosen to allow transport and to protect food from environmental interactions [21]. However, in recent years, a wide variety of packages and approaches have been used to interact with food to provide desirable effects; interaction with food may occur through compounds incorporated into the primary packaging material or by applying a layer between the package and the food. Such products, designed to perform various desirable functions other than providing an inert barrier, refer to GC and GT concepts, being called “active packaging” [22], ”reactive packaging” [23], “interactive packaging” [24] and “intelligent packaging” [25,26]. These materials have been developed to provide better quality, wholesome and safe foods and to limit packaging-related environmental pollution and disposal problems. Generally, they are smart systems used to extend food shelf life, while maintaining nutritional quality, inhibiting the growth of pathogenic and spoilage microorganisms, preventing and/or indicating the migration of contaminants, and displaying any package leaks present, thus ensuring food safety. Such products are particular primary packaging systems in the form of films (edible or not) and coatings; they are used to improve food shelf-life, quality, stability, safety and functionality, and also to prevent surface contamination, and to provide protection from physical, chemical and microbiological deterioration [27]. 

When a packaging material such as a film or a coating is food-derived and is eaten with the food, it is named “edible”; sometimes these two terms—edible films or coatings—are used as synonymous, but they are applied differently [28,29]. Edible films are considered as self-supporting structures produced separately and then either applied on food surface, between food components, or used to wrap food products; edible coatings are either applied to or formed directly onto food surface by dipping, spraying or panning [30]. Edible biopolymers like polysaccharides (e.g., starch, chitosan, and cellulose) or proteins (such as casein and gluten) can be used to obtain films or coatings [29]. 

Given that we are daily attacked by more and more aggressive microorganisms and viruses, a new challenge for the designers of food packages appears: is it possible to support human health through intelligent materials? The same question is current for the specialists in environment. How these goals could be achieved? We aim to provide a first answer to these questions, to show new points of view and to aggregate new ideas to obtain intelligent food packaging to help maintain human and environment health. In this context, sustainable starch-based primary packaging materials can be a reliable solution. Considering the starch-based films and coatings as reference, we propose to start “connecting-the-dots” between the concept of GP and two other green concepts—GC and GT—with a healthy integrative point of view; present work provides an overview of the recent studies regarding:

-Bioactive compounds (vitamins, polyphenols, essential oils, plant extracts, amino acids, etc.) and their role for the immune system;-GC methods to incorporate bioactive compounds into the starch matrix and their role for the human health;-GT, with emphasis on physical treatments with low environmental impact (UV irradiation, plasma and others) on the starch-based packaging with or without incorporated bioactive ingredients, together with a Green Nanotechnology (GN) approach in developing nanomaterials combined with starch for food packaging.

Ideas for designing future starch-based packaging are given, too, based on novel materials biocompatible with starch and on new concepts.

## 2. Starch-Based Films and Coatings

Among the biopolymers that can be used as food GP materials (Figure 2), starch is one of the most widely used due to its ability to produce films and coatings [31,32,33,34] and because of its biodegradability and renewability [35,36]. Starch can be found in different resources such as wheat, maize, potato, bean, rice and others. It is an important constituent of the human diet, being our major source of energy and representing 70–80% of human calories [37,38,39]. Starch’s basic unit is glucose, forming very complicated structures of two main polymers: amylose and amylopectin, whose ratio depends on the starch type [40].

Starch meets many of the expected features for an ideal candidate material for green packaging [26,41], being biodegradable, cheap, renewable, easy to process and safe for consumption. Another important characteristic is its compatibility with many other biopolymers, such as those indicated in Figure 2. 

The initial properties of starch can be modified/improved by chemical, physical, biochemical, and genetic actions and/or by their combination [42]. Starches whose properties have changed because of special processing are called modified starches. The modifications provide an improvement in the functionality of starches as thickeners, gelling agents, binders, adhesives, and film formers [37,43]. Chemically modified starches can be grouped into two categories, depending on the mode of action on the structure: starches whose branching degree is modified (acid-modified starches, oxidized starches and pregelatinized starches) and starches to which groups are introduced through copolymerization with other polymers or by chemical reactions (cross-linked starches, substituted starches) [44,45]. In order to obtain films, substituted starches are the most commonly used [46,47]; starch acetates give stable and transparent gels, forming resistant films after drying [48]. The introduction of acetate groups intensifies the stabilizing properties of starch and slows down the ageing processes [49,50,51]. The addition of plasticizers (commonly glycerol) overcomes film brittleness and improves the mechanical properties (flexibility and extensibility) [46]. Films can also be produced using plasticizers only; starch treated with plasticizers is called thermoplastic starch (TPS). Bioplasticizers, including small-sized and hydrophilic molecules like water, glycerol, monosaccharides (sucrose, fructose, glucose), amino acids, etc. [52], are commonly used because they are less toxic and are biodegradable [53]. In the presence of plasticizers and at high temperatures, starch exhibits thermoplastic properties [52]. The mechanical properties and water absorption of TPS depend on the degree of plasticization [54]. TPS is mostly used as coating [33,55]; its main actions are improving barrier properties (by reducing moisture and gas migration), suppressing physiological processes, delaying textural modifications, and improving the mechanical integrity of coated products [30]. TPS can form films as well.

The most commonly used methods for producing starch-based films are extrusion and casting; they are described in Section 4. Green chemical treatments of the starch-based films and coatings by incorporating bioactive compounds.

Starch shows shape memory. Shape-memory materials are advanced biopolymeric materials which undergo a phase transition between an initial temporary phase (leading to temporary shape) and a permanent phase (leading to permanent shape) [31], when exposed to a specific stimulus such as temperature, humidity, pH, etc. [56]. The shape memory of starch was first described by Chaunier and Lourdin in 2009 [57]. Further studies of Véchambre et al. [58] focused on the effect of moisture as a stimulus for the activation of shape recovery for amorphous starch-based materials with or without glycerol as a plasticizer; the efficient shape memory properties for the nonmodified starch have been highlighted in their study. Beilvert et al. demonstrated that shape memory of extruded potato starch with 20% glycerol triggered at 37 °C in water and can be used to develop biomedical devices [56]. 

Current research in the field of starch-based films is oriented in the following directions: improving the green production technology [32]; applying the GC principles by making the packaging material active or responsive [22,59]; increasing the mechanical properties [60], namely the tensile strength (TS) and elongation at break (Eb). TS expresses the ability of a material to withstand forces that tear it apart; Eb measures flexibility and stretchability (extensibility) prior to failure [61].

Figure 3 shows the evolution of publications (in Web of Science) on “starch food packaging,” in the last decade. The number of studies is not very high, showing that the research in this field is still at the beginning. There have been an increasing number of publications in the last years, whereas recently there has been a growing interest in using starch as a “green” natural resource for packaging, as well as “green” methods for materials synthesis and processing.

## 3. Plant-Derived Bioactive Compounds Promoting the Human Health

Considering starch-based packaging, a challenging issue is to make it a smart packaging material by identifying the adequate reason and technological solution. In the following section we focus on plant-derived bioactive compounds [62] which could interfere with the immunity and the microbiota/microbiome of humans, leading to multiple health-promoting effects like improved immunity, reduced inflammation, blood–brain barrier integrity, and the proper functioning of the gut [63]. Such multiple effects are linked to a variable extent to the immune system whose activation needs a marked level of the substrates which provide energy and are the precursors for the synthesis of new cells, protective molecules, and effector molecules [64].

Plant-derived bioactive compounds such as vitamins, polyphenols, essential oils, minerals, amino acids and lipids are all important for human health. Vitamins A, B, C, D and E are beneficial to the immune system. Vitamins A, C and E are responsible for maintaining the epithelial [65] and mucosal barrier in a good condition, which prevents eye, respiratory and digestive infections [66]. Moreover, these vitamins are also involved in the production of antibodies [65,66]. Vitamin A, due to the active metabolite retinoic acid, plays a key role in T-cell differentiation, T-cell migration in the tissues and a good development of the T-cell-dependent antibody response. It reduces inflammatory reactions and offers protection against infections [67]. Plus, Vitamin A regulates the innate cellular and humoral immune mediated response. Vitamin A also stimulates the white blood cells’ functions, and protects against the attack of infectious and carcinogenic agents [66]. Eggs, dairy products [68], meat [69], fish, beef liver [65], fruits and vegetables such as carrots, tomatoes (and tomato juice), spinach, oranges (and orange juice) [69], red potatoes and pumpkin [66] are rich sources of vitamin A.

B-Complex vitamins improve the immune response through their antioxidant effect and protect the body from diseases by stimulating the immunity [65]. Vitamin B6 has an important role in the growth and maturation of lymphocytes and the production of antibodies. In vitamin B6 deficiency, T-cell function is affected, and the size of the thymus is reduced. Vitamin B9 deficiency interferes with the cell division, and it influences the formation of blood cells in the bone marrow. In the human body, folic acid is a precursor of tetrahydrofolic acid—a compound that is used in the synthesis of proteins and nucleic acids. Vitamin B12 has a beneficial effect on immunity due to its role in cell division and maturation. Vitamin B12 deficiency inhibits the multiplication and maturation of white blood cells. Vitamin B6 is found in fruits and vegetables (potatoes, chickpeas, bananas) or animal sources (chicken breast, tuna). Folic acid is a constituent of bread, cereals, beans, peas and green leafy vegetables [65].

Vitamin C has beneficial effects on the cellular functions of the innate and adaptive immune system. It also possesses an immunomodulatory effect, and it is very effective antioxidant due to its capacity to donate electrons. Therefore, it protects some biomolecules (proteins, lipids, carbohydrates, and nucleic acids) against oxidants provided by the normal cellular metabolism or oxidants which come from hypoxia and/or exposure to toxins and pollutants (for example smoking). It stimulates the migration of neutrophils to the site of infection and increases phagocytosis. Vitamin C protects the host cell from excessive damage by increasing neutrophil apoptosis, macrophage elimination and decreasing neutrophil necrosis. Increasing the function of the immune cells by vitamin C leads to prevention and treatment of respiratory and systemic infections [70]. Vitamin C has also antiviral and antineoplastic effects. It stimulates the function of the leukocytes, especially the mobility of neutrophils and monocytes. In healthy adults and children, supplementation with vitamin C stimulates the chemotaxis of the neutrophils. Additionally, vitamin C supplementation stimulates the immune system by increasing the proliferation of T-cells as a response to infections. Therefore, there is an increase in the synthesis of immunoglobulins and cytokines [65]. Vitamin C is not synthesized in the human body [70] so it has to be provided through a diet rich in fresh fruits such as strawberry, citrus (lime, orange, lemon), kiwi, blackcurrant and papaya, and vegetables, such as broccoli, potato, tomato, green and red pepper [71], green leafy vegetables (spinach, kale), Brussels sprouts [65].

Vitamin D modulates innate antimicrobial and adaptive (acquired) immune responses. Vitamin D deficiency increases susceptibility to infections [72]. The vitamin D receptors are expressed on immune cells, especially on B cells, T cells, and antigen presenting cells, these being able to synthesize the active metabolite of vitamin D (calcitriol, 1,25-dihydroxyvitamin D3). Vitamin D receptors are generally localized in immature immune cells in the thymus and in mature CD8^+^T lymphocytes [65]. The dietary sources of vitamin D are cod liver oil [72], fish, meat, offal, eggs, and dairy products [73].

Vitamin E increases humoral and cellular immunity and phagocytic functions. Vitamin E has a greater effect in infectious diseases involving phagocytosis but is less effective in cases of cell-mediated immune defense. Studies show that a daily intake of 200 mg of vitamin E increases antibody responses to various vaccines in healthy subjects. Daily supplementation may rise the efficiency of the immune response to a specific antigen. Vitamin E also increases resistance to viral infections in the elderly people. Vitamin E is exclusively obtained from diet, and the richest sources of it are the vegetable oils (coconut, corn, palm, olives, peanuts, soy, wheat germ, sunflower, and saffron), nuts, seeds (sunflower), almonds, hazelnuts, vegetables with green leaves (spinach, broccoli), cereals, peanut butter, kiwi, mango and tomatoes [74].

Polyphenols are secondary plant metabolites, which have aromatic rings and hydroxyl groups in their chemical structure, and are found in various parts of plants, like leaves, fruits, seeds, wood and root [75]. The class of polyphenols includes flavonoids, tannins, phenolic acids, stilbenes [76], and anthocyanins [77]. Compared to volatile oils, polyphenols have some advantages: they are nonvolatile compounds, they do not have a strong odor and they are not lost by volatilization during storage [78]. They do exert multiple effects affecting genes’ expression implicated in the adaptive stress responses. More than 15,000 polyphenols have been identified in nature, some of them in fruits and vegetables, but also in cereals, oils, alcoholic and nonalcoholic beverages [77]. Grapefruit seed extract is a rich source of polyphenols (3.92%), especially flavonoids. Other important sources of polyphenols are green tea, grape seed, pomegranate, sea buckthorn, cloudberry, clove, passion fruit, turmeric extract [75] and red wine [76].

Polyphenols are traditionally extracted from plants by infusion, decoction, heat reflux, maceration, percolation or Soxhlet extraction, but these methods require a large amount of solvents and a very long extraction time. New technologies for the extraction of polyphenols are based on the use of microwaves, ultrasound, pulsed electric field and enzyme-assisted methods [79], and they can be studied using HPLC-MS [80].

Polyphenols have different beneficial effects for humans, including immunomodulatory [76], antioxidant and antimicrobial properties [81]. The food industry is increasingly interested in the development of innovative food packaging with antioxidant and antimicrobial properties [82]. The antioxidant activity of polyphenols is due to their ability to chelate metal ions and trapping reactive oxygen species [83]. Polyphenols also possess antimicrobial activity, but their mechanism of antimicrobial activity has not been yet completely elucidated [75]. The mechanism of antimicrobial action of polyphenols can be partially explained by changing the permeability of the bacterial cell membrane, altering the bacterial cell wall or by binding phenolic compounds to enzymes and altering intracellular functions [84].

Essential oils (EOs) are secondary plant metabolites that contain mixtures of lipophilic and volatile compounds with an aromatic odor. EOs have a complex composition, in which terpenes (mono- and sesquiterpenes) and their oxygenated derivatives predominate [85,86]. They are extracted from aromatic plants belonging to *Apiaceae (Umbelliferae), Asteraceae (Compositae), Cupressaceae, Hypericaceae, Lamiaceae, Lauraceae, Fabaceae, Liliaceae, Myrtaceae, Pinaceae, Piperaceae, Rosaceae, Rutaceae, Santalaceae, Zingiberaceae* and *Zygophyllaceae* families [87]. Essential oils predominate in various plant organs such as flowers (jasmine, rose, violet, lavender), the aerial part, buds (cloves), leaves (thyme, eucalyptus, sage), fruit (anise, star anise), twigs, bark (cinnamon), peel (citrus), seeds (cardamom), wood (sandalwood), rhizomes and roots (ginger) [88]. EOs are extracted from aromatic plants by conventional or modern methods. Conventional extraction methods are hydrodistillation, water steam entrainment, organic solvent extraction [89], Soxhlet extraction [90], and cold pressing extraction. Modern EOs extraction methods are supercritical fluid extraction, ultrasound extraction, and microwave extraction [89]. Qualitative and quantitative analysis of EOs can be performed by GC-MS or GC-FID [91]. 

Due to their antioxidant and antimicrobial activity, EOs are gaining increasing attention from the food industry [92], mostly for food preservation purposes [93]. *Origanum vulgare* L. volatile oil showed antimicrobial and antioxidant effects. The antioxidant effect of volatile oregano oil is due to rosmarinic acid [94]. Cinnamon volatile oil possesses an antimicrobial effect on fungi and foodborne pathogens [95]. Thymol and carvacrol are the main constituents of thyme volatile oil and they have antimicrobial and antioxidant effects. Due to their antimicrobial activity, when added to food products, thymol and carvacrol diminish the food spoilage by microorganisms [96]. Some researchers reported that thyme essential oil has antimicrobial effect against *Escherichia coli*, *Salmonella typhimurium*, *Bacillus cereus*, *Pseudomonas aeruginosa* and *Staphylococcus aureus* [97]. Volatile lemongrass oil showed antimicrobial effect against *S. aureus* and *E. coli* [98]. *Laurus nobilis* and *Myrtus communis* volatile oils have shown antibacterial effects against *S. aureus* [99]. Due to the already demonstrated antimicrobial effects of various plant-derived essential and volatile oils, it seems appropriate to use them for designing novel antimicrobial active packaging materials [100].

Minerals are inorganic compounds grouped in three classes, depending on the amount required by the body: macroelements (Ca, P, Na and Cl), microelements (Fe, Cu, Co, K, Mg, I, Zn, Mn, Mo, F, Cr, Se, S) and ultramicroelements (B, Si, As, Ni). Ca maintains the proper functioning of intracellular communication, the skeletal and muscular systems, it has a role in blood clotting, and participates in the activation of several enzymes [101]. Along with vitamins, minerals support the proper functioning of the immune system [66]. Iron is part of the structure of heme and it is involved in transporting oxygen to the cells [102]. Zinc contributes to the formation of bone marrow, and zinc deficiency can lead to decreased immune cell synthesis. Selenium stimulates immunity by preventing oxidative stress which leads to immunosuppression [64].

Amino acids (AAs) are the building blocks of proteins, and they are essential nutrients in our diet, playing an absolute critical role in many cellular and organismal functions. For example, tyrosine (Tyr) helps iron assimilation [103], and the amino acids like Leucine (Leu) and Glutamine (Gln) are important for efficient T-cells activation and for proliferative responses [104]. It was discovered that redox-active tyrosine (Tyr) and tryptophan (Trp) residues protect metalloenzymes from oxidative damage [105]. Another study emphasizes the role of Cysteine (Cys), Glutamine (Gln), Phenylalanine (Phe), Tryptophan (Trp) and Arginine (Arg) in T-cell function modulation [106]. The importance of amino acids in regulating immune responses through the following mechanisms was also highlighted: (1) the activation of T lymphocytes, B lymphocytes, natural killer cells and macrophages; (2) gene expression, cellular redox state, and lymphocyte proliferation; and (3) producing antibodies, cytokines and other cytotoxic substances [107]. Studies proved that a dietary supplementation of specific amino acids, especially to humans with infectious diseases, enhanced the immune status, thereby reducing morbidity and mortality. Moreover, Li et al. mentioned that Lysine (Lys) has a direct antiviral action, while Phillip Calder pointed out that branched-chain amino acids (BCAAs) are absolutely essential for lymphocyte responsiveness and to support other immune cell functions [64]. Whey is a veritable source of BCAAs (like leucine, isoleucine, and valine) that provide many health benefits including immunity enhancing and also repairing and rebuilding lean muscle tissues [108].

Lipids are a category of hydrophobic or amphiphilic molecules including: fatty acids, mono-, di- and triglycerides, animal and vegetable oils and fats, fat-soluble vitamins, sterols, phospholipids, lecithins, sphingolipids, natural resins, waxes [109]. They are important structural constituents, and fuel molecules providing the necessary energy for health promotion and supporting immunity (especially omega-3 fatty acids, conjugated linoleic acid), [110]. It is worth mentioning that activated immune cells have an anabolic metabolic profile, involving increased uptake and synthesis of fatty acids [111]. Fatty acids (FAs) are essential sources of energy and fundamental structural components of cells; they are hydrocarbon structures, and can be classified according to saturation degree, such as: saturated (which do not have a double bond), monounsaturated (one double bond, MUFAs) or polyunsaturated (two or more double bonds, PUFAs) fatty acids [112,113]. Between all FAs, it should be noted that monounsaturated fatty acids (MUFAs) like oleic acid (C18:1) and n-3 PUFAs play important role in immunity enhancement [114]. The Mediterranean diet (rich in fish and krill oils) is considered a suitable source of PUFAs. The oil extracted from Antarctic krill (*Euphausia superba*) is rich in the long-chain n-3 PUFAs eicosapentaenoic acid (EPA; C20:5 n3) and docosahexaenoic acid (DHA; C22:6 n3), which have immunomodulatory and anti-inflammatory properties [115]. Moreover, a recent report suggests that Omega-3 PUFAs: DHA and EPA, may improve COVID-19 associated mood symptoms via immunomodulation, since the fatty acids in question are essential for our brain and immune system, and they can only be obtained from proper diet [116]. Taken together, the incorporation of lipids into starch films could produce novel bioactive packaging materials. However, in contrast to proteins and polysaccharides, the lipids are not biopolymers, and they are unable to form self-supporting films for packaging. Therefore, lipids could be used either as coatings applied directly to food to provide a moisture barrier, or in combination with polysaccharides or proteins to create packaging films and coatings with low Water Vapor Permeability (WVP) and acting as a good barrier against gases such as O_2_ and CO_2_ [117]. It has been also suggested that the incorporation of lipids into films/coatings can improve cohesiveness, flexibility, hydrophobicity, and the moisture barrier properties of packaging [109]. 

Liposomes, lipid entities that mimic biomembranes, are used to build-up bionanomaterials with huge potential applications in the biomedical field and food industry (packaging, coatings and additives). Thus, improved biocompatibility, antioxidant, antimicrobial and anticancer activities were achieved by incorporation of active phytoingredients, drugs and phytonanometals into biomimicking lipid bilayers [118,119,120].

There has been reported a trend of the incorporation of total or fractionated plant extracts into food packaging since these substances contain complex matrixes of bioactive ingredients (polyphenols, vitamins, organic acids, proteins, etc.) with synergistic actions in the context of properties like augmented antioxidant, antimicrobial, anticancer properties, and immunity boosting. Thus, an active plant extract-based food packaging is expected to facilitate the relocation of bioactive compounds from the packaging material to the food, preventing the oxidative damage of the foodstuff, and avoiding the food spoilage by reducing microbial contamination. The antimicrobial action of plant extracts is also attributed to the presence of phenolic compounds that have hydroxyl groups (bound to a benzene ring) and a system of delocalized electrons [121]. Furthermore, the presence of plant extracts in starch films would reduce substantially the UV light transmission (200–3400 nm), due to the plant extracts contained aromatic compounds (like proteins, aromatic amino acids, polyphenols, etc.) that would act like an excellent UV barrier [122], offering good protection against photo-oxidation [123]. It is therefore likely that the plant-derived bioactive compounds, either alone or their extractlike combinations, could confer multiple advantages to the newly developed starch-based food packaging materials.

## 4. Green Chemical Treatments of the Starch-Based Films and Coatings by Incorporating Bioactive Compounds

GC was defined by the Environmental Protection Agency (EPA) as “the design, development, and implementation of chemical products and processes to reduce or eliminate the use and generation of substances hazardous to human health and the environment” [124]. The GC principles refer to several important sustainability aspects: (i) economy (low-cost, simplicity, rapidity); less energy and clean energy; the use of biocatalysts; (ii) safe materials (use of natural renewable resources and natural wastes, maximum recycling, the use of benign solvents like water, ethanol); (iii) waste reducing; (iv) minimizing and eliminating pollution, accidents, and any hazardous events [125,126].

Incorporating bioactive compounds into the starch-based films agrees with the GC aspects related to the economy, safe materials and minimizing the pollution. The literature research in the last few years on publications presenting starch-based films and coatings incorporating vitamins, polyphenols, essential oils, minerals, amino acids, peptides, proteins and enzymes, lipids and lipids-based nanostructures, together with plant extracts containing bioactive compounds, is presented below.

Different techniques, such as casting and extrusion, can be used to produce starch-based edible films incorporating biocompounds. The extrusion process of edible film production does not use liquid solvents, such as water or alcohol; whereas for the dry process, heat is applied to the film-forming materials in order to increase the temperature needed to overcome the melting point of the film-forming starchy materials, thus causing their flow [127]. Extrusion has the advantage of being a low cost, continuous and versatile production system; this technique can be used for large-scale production [128,129]. 

The casting process uses solvents for the dispersion of film-forming materials, followed by drying to remove the solvent and form a film structure. The film-forming dispersions should be applied to flat surfaces using a sprayer, spreader, or dipping roller, and dried to eliminate the solvent, forming a film structure [127,130]. Casting has been extensively used in laboratory studies because it does not require specific equipment and consumes lesser amounts of raw materials; many researches refer to films obtained by casting [131,132,133,134,135,136].

### 4.1. Vitamins Incorporation into Starchy Films and Coatings

The research in incorporating vitamins into starch-based films and coatings is just beginning. The incorporation of the vitamins which possess antioxidant effect may be done during the manufacturing process because they protect against the film’s oxidation [137]. The most commonly used method for incorporating vitamins into starch film is the solvent casting method [131,132]. Fakhouri et al. described the manufacturing process of the starch films incorporating cranberry powder and evaluated their thermal, microstructural, mechanical, sensorial characteristics and ascorbic acid content. The ascorbic acid contained in cranberry powder improves the sensorial properties of the films, making them more attractive to tasters. The film acts as a protection for ascorbic acid, preventing its degradation by light, heat or oxidation [131].

Other researchers evaluated the properties of rice starch film with ascorbic acid [132].

Ascorbic acid film obtained by incorporating cranberry powder into the starch matrix of *Maranta arundinaceae* L. can be used in packaging fruit stripes as a source of nutrients or as coating for sushi [131].

### 4.2. Polyphenols

They are many researches on adding polyphenols into biopolymers-based films. The alginate films with polyphenols from tea showed antioxidant and anti-inflammatory effects [78]. Polyphenols from apple peel were reported to have antioxidant activity [138]. According to Wu et al., tea polyphenol addition into a bioactive film manufactured from pomelo peel flour led to an increase in the antimicrobial and antioxidant properties of the films. Additionally, the decrease in the transmittance, moisture content and Eb were observed in this case. The films with tea polyphenols showed an improvement in the water barrier due to their compact structure [82]. Riaz et al. showed that incorporating apple peel polyphenols into chitosan-based films led to an increase in the opacity, thickness, solubility, swelling ratio and density of the films. On the other hand, they observed that WVP and moisture content were lower. An increase in the antioxidant and antimicrobial effects of the chitosan-based films with apple peel polyphenols was observed [138]. Feng et al. reported that addition of tea polyphenols into starch matrix led to a better antioxidant activity of the films, and they also observed that the starch-based films with tea polyphenols have antibacterial effect against *S. aureus* and *E. coli* [83]. 

Table 1 presents a summary of some recent studies which reported polyphenols’ incorporation in different films used as food packaging and their applications.

### 4.3. Essential Oils

The addition of EOs to food packaging can increase the shelf life and quality of food by protecting the consumer from the harmful effects of oxidative stress and microorganisms. The introduction of EOs in food packaging may change the characteristics of the films used as packaging and this influences the acceptance of the product by the consumer. The frequent use of food packaging with essential oils can lead to hypersensitivity reactions [144].

Nisar et al. investigated the properties of citrus pectin films with clove bud EO. This study showed that the dispersion of clove oil in the films led to an increase in the water barrier and opacity of the films. The films with clove volatile oil exhibited antioxidant activity. Clove bud EO incorporated in citrus pectin films showed inhibitory effect against some foodborne pathogens (*S. aureus*, *E. coli*, *L. monocytogenes*) [145]. In another study, thyme volatile oil microcapsules were incorporated in corn starch films made by casting method, then the film was used for conserving mango fruits. The study showed a higher stability of the microcapsules when added into the film, but films with incorporated microcapsules are more sensible to temperature than films without microcapsules. Films with thyme EO microcapsules were more opaque, thicker, more tensile resistant and more soluble in water. These films with thyme essential oil microcapsules showed antimicrobial activity against *Botryodiplodia theobromae* Pat. and *Colletotrichum gloeosporioides* Penz. [133]. Eugenol added in the form of microcapsules in corn starch films has proven antioxidant activity and these films can be used to preserve sunflower oil to prevent its oxidation over time [146]. Other researchers analyzed the film of cassava starch/polyvinyl alcohol/sodium alginate with volatile lemongrass oil and copaiba used in the preservation of lettuce by refrigeration [147]. Another study showed that an emulsion of volatile lemongrass oil incorporated in a cassava starch film improved optical and mechanical properties of the cassava starch film [98].

EOs can be easily fabricated as microcapsules and nanoparticles, which increases their stability and solubility. Hence, EOs are considered as the most usable additives in future functional foods [148].

### 4.4. Minerals

Minerals form complex starch structures through coordinative bonds that are established between the hydroxyl groups of starch and the metal ion. During the heating of the mixture of starch and minerals, an increase in viscosity was observed; therefore the time required for the drying of the films is prolonged. Iron and manganese ions give a yellow-brown color to starch films. Manganese and zinc ions increase the ability of films to retain water. It has been observed that the incorporation of minerals into the starch matrix leads to a decrease in water vapor pressure and an increase in tensile strength (TS) [102].

The incorporation of metals, metal oxides or organic compounds into films can increase their antimicrobial activity. Magnesium oxide (MgO), zinc oxide (ZnO), silicon dioxide (SiO_2_) and titanium dioxide (TiO_2_) have antimicrobial activity, but they also protect against UV radiation. Zinc oxide and magnesium oxide are the most suitable and safest mineral compounds used for food packaging [149].

Due to its broad antimicrobial spectrum, its stability to heating and its safety, ZnO is a compound of interest for the food packaging industry [150]. Zinc has different roles in the human body. It is an essential trace element which supports the immune system, helping to form bones and healing wounds [151].

Moreover, the antimicrobial properties of silver and zinc oxide in nanoparticulate forms are exploited in development of packaging materials, as described further in the Section 6 (Nanotechnology in starch food packaging).

### 4.5. Amino Acids, Peptides, Proteins and Enzymes in Starch-Based Food Packaging

Charge-carrying amino acids (Lys, glutamic acid (Glu), aspartic acid (Asp) and arginine (Arg)) could modify the physicochemical properties of starch gels (significantly decreased the swelling power, solubility, gel strength, and light transmittance), while neutral amino acids (methionine (Met) and phenylalanine (Phe)) did not induce modifications [152].

Proteins are polymers of AAs linked together by peptide bonds in a specific spatial architecture providing unique biological function. These valuable biopolymers are exploited in developing “green” starch-based packaging materials. The most commonly used proteins in food industry are those derived from milk, whey, eggs, soybean, wheat, corn, sunflower, peanut, cottonseed, rice, fish, silk, etc. [31,109]. 

Whey proteins–starch films embedded with rambutan peel extract and cinnamon oil exhibited antioxidant and antibacterial activities, and also improved mechanical and barrier properties [153]. 

The films based on proteins provide a barrier for O_2_ and CO_2_ but do not resist water diffusion. Moreover, films made of proteins and carbohydrates are excellent barriers to O_2_, since the formation of ordered H-bonded networks between the two types of biopolymers [109]. The addition of plasticizers, such as glycerol or polyethylene glycol, improves protein film flexibility, whereas WVP can be decreased by adding hydrophobic materials such as oleic acid or beeswax. Furthermore, the introduction of plant-derived antioxidants into edible protein films is an emerging trend today [109] to extend food shelf life. On the other hand, the inclusion of nanofillers such as AgNPs, nanocellulose, montmorillonite improves mechanical strength, thermal stability, and water and oxygen barrier properties [154]. 

Starch–proteins blends processed in the presence of plasticizers lead to obtaining biodegradable plastics with enhanced properties. Compared with animal proteins, plant protein sources are preferred due to their low cost, wide availability, and renewability [152].

Some studies demonstrated that starch-protein interaction modified the film’s surface energy, enhancing surface hydrophobicity. Quéré stated that wettability of hydrophilic surfaces increased with increasing roughness [155]. The incorporation of starch in protein films resulted in increase in smoothness due to strong hydrogen bonding of polymers, thus reducing surface roughness and resulting in increased contact angle [153].

Leroy’s research team [156] prepared zein–starch films with reduced sensitivity to water by using glycerol and 1-butyl-3-methylimidazolium chloride as plasticizers.

One of the most commonly used proteins for food packaging film production is gelatin, an animal protein with excellent film-forming properties, obtained by hydrolysis of collagen, which is generated during animal slaughtering and processing. The main sources of gelatin are: fish, bovine, and porcine [157]. In addition, gelatin is a very abundant and fully digestible protein, containing nearly all the essential amino acids, excepting tryptophan, and gelatin-based food packaging films have barrier properties against oxygen, lipids, UV light, and heat sealability [116,158]. Tosati et al. [159] obtained edible coatings based on gelatin and turmeric residues (containing starch, fibers and curcuminoids), which possess attractive features: they are flexible and malleable and have antimicrobial effect. Very recently, Gopal highlighted in his review [160], that consuming turmeric can help in boosting our immunity in the present scenario, when this devastating pandemic situation (SARS-CoV-2) has caused more than 2 billion deaths.

In addition, protein-based films and coatings are biodegradable and compostable, and their degradation provides a source of nitrogen, which acts as a fertilizer. Moreover, protein hydrolysis results in the formation of peptides, which are short chains of AAs linked together by peptide bonds; these biomolecules are bioactive and possess health benefits (antioxidant, antimicrobial or antihypertensive properties) [161]. An interesting class of peptides is represented by bacteriocins, which are small bacterial peptides with antimicrobial activity. Nisin, pediocin and natamycin are the most commonly used bacteriocins in starch film preparation [123]. 

Resa and coworkers [162] developed antimicrobial starch-nisin and starch-natamycin films against *Listeria innocua* and/or *Saccharomyces cerevisiae*.

Recently, Meira et al. [163] dispersed pediocin and nisin in starch films, thus obtaining active packaging materials with antimicrobial activity against *Listeria monocytogenes* and *Clostridium perfringens*. 

Enzymes represent a group of proteins which catalyze the chemical processes in living systems, which also have application in food packaging. Lysozyme, an important class of enzymes, present in many foods, such as milk and eggs, has a hydrophilic monopeptide chain and inhibit bacterial infections (especially those caused by Gram-positive bacteria), by hydrolyzing the peptidoglycans which are the main bacterial cell wall components of Gram(+) bacteria, causing loss of intracellular content and, finally, bacterial death [123,164]. Bhatia and Bharti [165] obtained antimicrobial starch-based active food packaging film by incorporating nisin and lysozyme as natural biocides.

### 4.6. Lipids and Lipid-Based Nanostructures in Starch Packaging Systems

The packaging starch films are generally effective oxygen hinderers at intermediate to low humidity, but they are a poor water vapor barrier because of the starch hydrophilic properties [166]. Lipids are hydrophobic or amphipathic biomolecule, their addition to the starch films and starch-based composites improves the water barrier properties. Moreover, lipids interplay with immune regulation, providing energy to T-cells but may also regulate T-cell function by an immune checkpoint such as PD-1 [111].

On the other hand, the addition of lipids into starch packaging materials improves the mechanical barrier and optical properties of the film due to the change in inner structure and film surface [167]. Therefore, examples of such films are as follows: (1) epoxidized soybean oil/TPS films with high Young’s modulus and TS [152]; (2) cassava starch/glycerol/carnauba wax/stearic acid with good barrier and mechanical properties [152]; (3) films consisting of starch and fatty acids and phenolics from basil seeds, with increased hydrophobicity [168]; (4) chitosan/lauric acid/starch film with antimicrobial ability against *B. subtilis* and *E. coli* more effective than chitosan alone [169,170]. 

Another study [171] showed that the incorporation of saturated fatty acids—caproic, lauric and stearic—did not change the nature of the chemical bonds among components in the starch blends according to the infrared, Raman spectroscopic techniques, and thermal analysis. Kapusniak and Siemion [172] developed a “green” method to obtain starch–linoleic acid blends, in which potato starch was esterified with linoleic acid by a thermal reaction, and found that as the degree of starch esterification increased, the water binding capacity of the films decreased, and the susceptibility of the esters to α-amylolysis slightly decreased [172].

Slavutsky and Bertuzzi showed that coating of lipid nanolayer (sunflower oil) with hydrophilic film (starch) caused an increase in TS and also low water diffusion coefficients since the structure and the hydrophobic nature of the lipids and oils restrict the migration of gas and vapors [173].

Natural oils, especially vegetable oils, are currently used as green plasticizers for the compatibilization of polymer blends to obtain novel green materials for food packaging applications. Plant-derived oils are interesting due to their triglyceride structure, consisting of a glycerol backbone which is chemically bonded to different fatty acids through ester bonds. To give desired functionality to a vegetable oil, different chemical modifications are required: epoxidation, hydroxylation, maleinization, acrylation, etc. Thus, Gonzalez et al. reported that the addition of maleinized linseed oil to poly(lactic acid)/diatomaceous earth (PLA/DE) composites resulted in an increase in thermal stability, chain mobility, and Eb, together with a decrease in rigidity and TS [174]. These findings could be exploited for active packaging applications.

Lipid-based nanostructures (lipid nanoemulsions, liposomes, solid lipid nanoparticles and nanostructured lipid carrier systems [175]) are another interesting application of lipids into active food packaging. This nanotechnological approach allows the migration of the active ingredients from lipid nanocarriers to the food matrix or onto the food surface.

Vegetable-based lipid nanoformulations are a current research topic in the design of active packaging materials. Exploiting natural resources in combination with soft nanotechnology gives rise to valuable products with improved biological performances. The research team of Lacatusu [176] developed high antioxidant lipid nanocarriers based on hempseed and amaranth oils in association with lipophilic plant extract enriched in carotenoids originated from Marigold plant (*Tagetes patula*). 

In last few decades, liposomes have gained more attention in many fields (medicine, pharmacology, cosmetics, nanotechnology, food industry), due to their fascinating properties: self-healing, ability to carry both hydrophilic and hydrophobic molecules, capacity to improve the bioactivities of the substances they carry, great resemblance to cell membranes. Liposomes are self-closed lipid vesicles containing one or more concentric phospholipid bilayers that include aqueous compartments. They are nontoxic, biodegradable, biocompatible, nonimmunogenic, and they can entrap hydrophobic (within lipid bilayers), hydrophilic (in the aqueous regions) and amphiphilic (at the aqueous–lipid interface) molecules [177,178]. The encapsulation of vegetal extracts in the liposomes produced more antioxidant activity than the same extract alone [170]. Cui et al. reported the preparation of SiO_2_-eugenol liposomes with antioxidant activity and great potential as a food packaging material [179].

Liposomes have also been explored as antimicrobial incorporation systems for the development of new biocidal packaging materials [180]. Thus, some studies reported that nisin, an antimicrobial peptide produced by strains of *Lactococcus lactis*, entrapped in phosphatidylcholine liposomes, was incorporated to biopolymer-based films (of hydroxyethyl cellulose, gelatin or casein), resulting in active packaging materials that exhibit antimicrobial activity against *L. monocytogenes* [181,182,183].

### 4.7. Vegetal Extracts in Starch Food Packaging

The incorporation of aqueous rosemary (*Rosmarinus officinalis*) extracts in edible and biodegradable cassava starch film (plasticized with glycerol), and the effects on the physicochemical properties of developed film, were firstly reported by Piñeros-Hernandez et al. [122]. The prepared active films showed a significant increase in the contact angle values since of hydrophobicity of rosmarinic and carnosic acid, the main bioactive compounds present in aqueous rosemary extract. Moreover, these active starch films exhibited high elastic modulus values, high antioxidant activity and UV-blocking properties, plus a high biodegradation extent.

Thyme (*Thymus serpyllum* L.) leaves’ aqueous extract, a rich source of polyphenols, was inserted into chitosan/starch film formulations to provide edible films with remarkable antioxidant activity and improved mechanical properties (TS), with great potential in food preservation [184].

Furthermore, Luchese et al. incorporated dried and grinded blueberry (*Vaccinium corymbosum* L.) peels, and sorbitol into cassava starch, resulting in starch films with low water solubility and enhanced UV-light barrier properties [185]. This is a good idea to minimize waste generation, and to give value to the agrowastes from the blueberry juice industry, by incorporating them into fully biobased films.

Rosehip (*Rosa canina* L.) extract, rich in antioxidants (such as ascorbic acid and carotenoids), was added to rye starch films and improved the mechanical and chemical properties of these films (decrease in TS, increase in Eb and in flexibility, and improving the light barrier properties) [186]. These starch–rosehip films presented antioxidant activity which was exploited as wrapping film to inhibit lipid oxidation in chicken breasts. 

Another study [187] reported the incorporation of rice straw extract into potato starch films by melt blending and compression-molding. The developed films exhibited improved barrier, mechanical and thermal properties, and antioxidant and antimicrobial activities, also being a promising candidate for biodegradable antioxidant packaging material. 

The research group of Baek reported for the first time the use of cowpea (*Vigna unguiculata*) for starch-based films’ preparation [188]. They incorporated maqui berry (*Aristotelia chilensis*) extract (rich in antioxidant biomolecules such as delphinidin derivatives) into cowpea starch films and demonstrated their applicability in the food industry for salmon packaging.

Chollakup et al. [153] reported the preparation of active blending films consisting of cassava starch, whey protein isolate, rambutan (*Nephelium lappaceum*) peel ethanolic extract and cinnamon (*Cinnamomum zeylanicum*) oil. These starch-based films presented good antioxidant activity attributed to the presence of phenolic compounds such as corilagin, geraniin, and ellagic acid [189], arising from rambutan extract. The films also showed high antibacterial action against *Bacillus cereus*, *Staphylococcus aureus* and *Escherichia coli*. It should be mentioned that corilagin (β-1-O-galloyl-3,6-^®^-hexahydroxydiphenoyl-d-glucose) is ydrolysableble tannin (an ellagitannin) with a wide therapeutic spectrum, such as antioxidant, anti-inflammatory, antimicrobial, hepatoprotective, antihypertensive, antidiabetic, and antitumor activities [190]. Moreover, the hydrolytic products of corilagin: ellagic acid and gallic acid, possess anticancer activity, being involved in stimulating the cellular immune response [191,192].

In addition, tannins act as immunomodulatory agents in the battle against infectious diseases, being effective against the methicillin-resistant *Staphylococcus aureus*, and also have antiviral activity against HIV (human immunodeficiency virus) and HSV-2 (herpes simplex virus 2) [193].

*Eugenia uniflora* L. (named also Brazilian cherry, Surinam cherry or Pitanga) aqueous extract was incorporated in cassava starch (CS) and chitosan (CH) blend films, which resulted in effective antioxidant and antifungal food packaging material [194] with a UV barrier effect attributed to the phenolic components derived from pitanga extract.

This innovative food packaging concept based on vegetal extracts should be the center of attention of researchers since the incorporation of plant extracts into starch films is a promising method to prevent or reduce food quality deterioration, thus preserving and extending food shelf-life by providing improved physical, mechanical, structural and barrier properties [195].

By taking into account all aspects discussed here, the main actions of starch-based primary packaging in the form of edible films and coatings incorporating the investigated bioactive compounds and the plant extracts containing these biomaterials can be grouped as:

-Actions sustaining consumers’ health and safety: boosting the immune system, antimicrobial and antioxidant activity; -Actions on the packaged products: increasing their shelf life, improving of the mechanical and sensorial properties.

An overview on this approach is given in Figure 4.

## 5. Application of Green Physical Treatments on Starch and Starch-Based Films

Even though the chemical methods are the most widely spread starch modification methods [41], in the last two decades physical modification techniques are in demand in the food industry, especially due to their residual-free process, which means the resulting starch is not labeled as modified starch [196]. Currently, starch physical treatments can be divided into thermal and nonthermal treatments [196]. As high temperatures induce the deformation of starch granules [197] we will include only the following nonthermal treatments: UV, electron beam and gamma irradiation and plasma. We name these treatment methods as nonthermal since heat is not intentionally applied, even though heat may be a consequence of the applied treatment.

Physical methods used for starch treatments involve the application of various intensities of heat, pressure, and radiation [36]. As mentioned earlier, these methods can be grouped into two categories: thermal and nonthermal methods. Thermal treatments refer to the pregelatinization of starches, heat-moisture method, annealing microwave heating, osmotic pressure heating and dry heating of starch, whereas nonthermal treatments include ultrasound, milling, ultra-high-pressure method, cold plasma, pulsed electric field, high-pressure homogenization, freezing and thawing, gamma irradiation, ultraviolet (UV) irradiation and ozone treatment. Nowadays, the most commonly used physical starch treatments are thermal treatments [196]; however, in this review we will focus on novel processing methods. This is due to the fact that recently there is a high demand for industrial scalable emerging ecofriendly, green technologies that do not generate toxic waste or residual chemicals and react with starch molecules at room temperature, requiring no specific temperature conditions for starch treatment.

New research has also been focused on utilizing green physical methods to improve physical and chemical properties of the starch-based bioplastics [198]. We highlight that in this review “green” physical methods refer here to physical modification techniques that do not involve added reagents, are user friendly and are industrially scalable at lower costs [199,200].

Irradiation is the process of radiation energy that is exposed to a certain type of the polymer or starch to achieve desirable changes [201]. The most common radiation treatments are UV radiation, gamma radiation and electron beam [202]. UV radiation is one of the physical methods that is environmentally friendly technology and may modify the physical, chemical, or biological characteristics of the products [203]. Irradiation with ionizing radiation of the polymeric materials can lead to the formation of highly reactive intermediates, free radicals, ions, and excited states. These intermediates follow many rapid reaction pathways leading to disproportion, hydrogen absorption, adjustments and/or the formation of a new polymer chain bond, thus altering the final structure of the network structure [204]. Electron beam (EB) irradiation is a low-cost technology and environmentally friendly, without any cause or use of polluting agents, catalysts, or generation of undesirable wastes [205]. 

Gamma irradiation is the process of using an isotope from Cobalt-60 and results in a high energy photon [206]. Gamma irradiation is very penetrative compared to the other forms of radiation [207] and causes chemical changes in macromolecules; the main process that occurs is macromolecules’ degradation with subsequent oxidation by atmospheric oxygen, leading to the formation of the carbonyl and derivatives. Radiation contact with matter contributes to the creation of positive ions and excited molecules to create radicals. A polymeric radical and a hydrogen atom are produced after each excitation–ionization—some of those hydrogen atoms released in the immediate vicinity with substantial kinetic energy, creating secondary polymeric radicals, a pair of adjacent radicals, one formed by radiation and other by abstraction, which can then be easily cross-linked [208].

Cold plasma techniques are based on plasma and the term “plasma” has been used since 1928 to define the fourth out of five states of matter [209]. A plasma is an ionized gas consisting of electrons, ions, free radicals, excited state, and neutral molecules [200]. Human-made plasmas or laboratory plasmas can be divided into high-temperature and low-temperature plasmas, or fusion plasmas and gas discharges, respectively [210]. Gas discharges are the subject of the present review because they are suitable to treat temperature sensible materials, including starch. Furthermore, gas discharge plasmas can be divided into “local thermal equilibrium” and “nonlocal thermal equilibrium” plasmas, depending on the temperature of the contained plasma species. In both cases, pressure as well as the discharge length play a crucial role. On top of this, plasma effects are influenced by gas composition, humidity, power/applied voltage, and the surrounding phase [211].

### 5.1. UV Radiation

According to [212], UV radiation improves the mechanical properties of starch. The high energy of UV irradiation would have a high effect on starch properties and result in creating free radicals that would generate crosslinks between the structural components [213].

Nawapat and Thawien [214] analyzed the influence of UV treatment and benzoate (used as photosensitizer) on rice starch biofilms. The TS of the rice starch films was affected by the increase in photosensitizer, whereas the increase in the photosensitizer did not show the same result in Eb. Moreover, the UV-treated rice starch films showed lower WVP than untreated rice when the amount of the photosensitizer increased. 

At the treatment of a sago starch/PVA film with UV, TS increased with the rise in the radiation intensity; however, the TS started to decrease after reaching a maximum value, this behavior being due to the polymer chain cross-linking density. The TS value decreased when the concentration of starch increased; the cross-linking density increased with increased doses of UV radiation, which led to higher Eb [215]. Monomers play an important role in improving the physical properties of the UV cured films by creating free radicals; the TS and Eb values will increase with increase in the concentration of the monomers, as Khan et al. showed [215].

Moreover, UV radiation affects the transparency of the starch films; transparency decreases under UV-C (200–280 nm) treatment [216], this phenomenon being observed in other studies, too [214,217].

The microstructure of the starch films treated with different wavelength (UV-A 315–400 nm, UV-C 280–300 nm) does not show cracks on the surface for long time UV exposure (more than 6 h) [216].

Due to starch’s hydrophilic structure, starchy films have poor barrier properties [218]. The WVP ability of the starch-based films decreases after exposure to UV, compared with the untreated samples. Regarding the mechanical properties of the treated and untreated starch films, TS in the UV treated films show a decrease with an increase in the exposure time and the wavelength; the untreated films have higher tensile strength values [216].

Even though starch films are excellent barriers to oxygen [219], their mechanical properties are somewhat limited due to their high sensitivity to moisture [220]. According to [221], the TPS produced with UV treatment and the addition of boric acid shows good mechanical and barrier properties. The percentage of the sensitizer is important, its increase giving a decrease in TS.

### 5.2. Ionizing Radiation

#### 5.2.1. Electron Beam (EB) Irradiation

To overcome the concerns caused by the limitation of the native starch, modifications have been made to improve its functions and properties [222]. Electron beam (EB) processing has been reported to be effective in inducing physicochemical changes in starch structure, for example increasing solubility and free acidity, decreasing the swelling capacity, viscosity, pasting properties, molecular weight and degree of polymerization [205,223,224]. However, as stated by Uehara and Mastro [225], TS increases as radiation doses increase and turns to yellow color. According to other research [226], the physicochemical properties of corn starch were modified by treating them with accelerated EB doses (0, 10, 20, 30, 40, 50 kGy). The acidity values of the EB-treated samples were increased while the received dose increased, which may be due to the fragmentation of starch molecules and the formation of compounds containing the carboxyl group. In addition, the EB treatment in the presence of oxygen caused the formation of free radicals, aldehydes, ketones or other forms of polysaccharide degradation products, all of which led to an increase in starch acidity [227].

The physicochemical properties of sago starch have been impacted by the EB irradiation. Amylose and amylopectin degradation resulted in a shift in physicochemical properties. When the irradiation dose was increased, peak viscosity, yellowness, redness, solubility, and free acidity were increased, whereas the gel pressure, swelling power and pH were reduced. The intrinsic viscosity was reduced by the increase in the EB irradiation [228].

Zhou et al. [229] analyzed the effect of EB irradiation on waxy maize starch structure, degradation, and thermal and mechanical properties of the films. The results showed that the proportion of linear chains increases with a sufficient dose of irradiation, the average molecular weight of starch molecules decreases, and the mechanical properties and solubility of the starch films improve. An EB irradiation at doses of 10 kGy moderately degrades starch and increases the number of linear chains, resulting in better TS and Eb in starch films, together with an improved solubility [229].

#### 5.2.2. Gamma Irradiation

Zhai, Yoshii, and Kume [230] reported that TS of the starch-based plastic sheets increases when the irradiation dose ranges between 30-70 kGy and decreased when the irradiation dose is higher than 120 kGy. Moreover, according to Naime et al. [231], the increase in the irradiation dose will lead to an decrease in Eb and an increase in the water absorption [61]. Gamma irradiation is found to improve mechanical and barrier properties of processed starch biofilms [232].

Irradiated corn starch was used for casting. Irradiation doses of 10, 20, 30 and 40 kGy resulted in a positive increase in the films’ TS. The water vapor permeability of the film decreased. The best properties of the films with the improved characteristics were shown at the irradiation doses of 30 kGy [134].

During irradiation, the chemical reaction leads to the creation of an intact network structure in PVA/chitosan (CS) films. Radiation-induced crosslinked reaction increases the TS of PVA/CS films. However, under high energy irradiation, three forms of species are formed and may become entangled in polymers, ionic species, radicals, and peroxides. Postirradiation can be caused by both radicals and peroxides, and the various active centers can lead to various chemical transformations such as crosslinking and degradation [233].

The gamma irradiation of cassava starch gave foams developing stable formulations at doses of 3 kGy, 6 kGy, 12 kGy and 25 kGy; the mechanical (compression resistance) and barrier (water solubility) properties increased compared to foams obtained with nonirradiated starch. With the increase in the irradiation dose, the absorbed amount of the water increased, whereas the mechanical and barrier tests did not follow the same trend [231].

### 5.3. Other Green Physical Treatments

#### Plasma Treatment of Starch

Regarding starch modification by plasma, attention must be paid when choosing from the multitude of the existing plasma types [234] to achieve a specific outcome for starch processing. For instance, the first plasma starch modifications were done at the beginning of the 2000s [235,236,237,238,239,240] and belong to the “nonlocal thermal equilibrium” plasma category, or the so-called cold plasmas. Since then, it was found that starch can be modified through various mechanisms: crosslinking or grafting, depolymerization, plasma etching and surface functionalization with functional groups [200].

A multitude of review articles were published in recent years that address cold plasma treatments of starch [36,37,196,199,200,241,242,243,244]. However, this is an emerging research field and cold plasma modified starch has not yet been studied in real life food systems, such as coatings or packaging films. Therefore, we present a literature review focused on atmospheric pressure cold plasma treatments, which is a plasma-based method suitable for industrial use, technology transfer, and has lower maintenance costs [245,246]. This category of plasmas is a particular “green” physical method that has the potential to produce “green” starch products [247], with real chances to turn the resulting plasma modified starch into generally regarded as safe (GRAS) for consumption [200]. An exemplification of the differences in cold plasma set-ups is presented in Figure 5, where schematic drawings representing low-pressure and atmospheric pressure plasmas used for polymeric material treatments are shown [248,249].

It can easily be seen that the atmospheric pressure plasma method requires a less complicated set-up, with fewer components, and is an instant ready-to-use method. On top of the beforementioned advantages, atmospheric pressure cold plasma treatments are seen as innovative techniques for food processing [211]. Therefore, Table 2 summarizes the original research articles and patents literature regarding atmospheric pressure cold plasma treatments applied for starch modification and treatment that have a potential use in the food industry.

## 6. Nanotechnology in Starch Food Packaging

Starch presents some limitations attributed to its hydrophilic character, and like any biopolymer, starch’s mechanical properties are poor, and it is brittle [38]. The nanotechnological breakthrough, especially Green Nanotechnology (GN), tries to solve these problems and to greatly improve the properties of starch-based films. In the last two decades, another interdisciplinary research area was intensively developed, polymer nanotechnology, which uses nanocomposite systems based on polymers for food packaging applications. These nanocomposites, in which the dispersed phase is nanostructured, represent a new class of materials called nanopackaging [262]. The nanosystems used in food packaging can be classified in three categories, depending on their size and shape, as follows [262]:

Particulated 3D nanoreinforcements (nanoparticles and nanocrystals);Fibrillated 2D nanoreinforcements (nanofibers, nanotubes);Laminated 1D nanoreinforcements (nanoclays).

The main nanoreinforcers used in packaging include: natural nanoparticles (nanochitin, nanochitosan, nanostarch, nanocellulose, nanoalginate, liposomes), nanoclays (montmorillonite), metallic nanoparticles (MNPs, where M = Ag, Au, Se, etc.), metal oxide nanoparticles, MONPs (ZnONPs, TiO_2_NPs, MgONPs), nanosilica (SiO_2_NPs), carbon nanotubes/fullerenes, etc. [174,262,263,264]. The combination, in an original manner, of these nanomaterials with biopolymeric matrix, gives rise to food nanopackaging systems with improved features (strength; stiffness; flexibility; temperature/moisture stability; heat resistance; barrier protection to light, gases and water; bioactivities: antimicrobial, antioxidant and anticancer properties), thus providing protection and preservation of the content, and also the reduction to zero of any critical interaction with the food matrices and with human health [262]. The use of fillers with at least one nanoscale dimension favors strong interactions between nanofillers (nanoreinforcements) and the matrix, attributed to high surface/volume ratio [262], which allow exposure of more surface atoms compared to their microscale counterparts, resulting in interesting optical, catalytic and other reactive properties [265].

We will further mention some studies conducted in the last two decades. The permeability to oxygen was considerably decreased when sago starch/bovine gelatin bionanocomposite films were filled with zinc oxide nanorods (up to 5 wt% total solid), and plasticized with sorbitol/glycerol (3:1) [266].

A group of researchers incorporated ZnO nanoparticles in different films made from agar, carrageenan and carboxymethyl cellulose. They observed that nanoparticles were uniformly distributed into the matrix and the colors of the films were modified by ZnO nanoparticles. The incorporation of ZnO nanoparticles into the films led to increased hydrophobicity, UV barrier, Eb value, moisture content and thermal stability of the films, whilst the water vapor barrier and TS were lower. The films with ZnO nanoparticles showed antimicrobial effects against *L. monocytogenes* and *E.coli* [150].

A significant decrease in the moisture sensitivity of the biopolymer matrix was achieved by Hietala et al. [267], by incorporating different amounts of cellulose nanofibers in a TPS matrix.

Through a casting-evaporation method, Salaberria and coworkers [135] obtained TPS-based films containing nanochitin with different morphologies (nanofibers and nanocrystals), with decreased water vapor transmission values.

A simple casting method was employed by Gonzalez and Igarzabal to develop soy protein films reinforced with starch nanocrystals (SNC) at different concentrations. The presence of SNC as nanofillers was found to decrease the affinity of the films for water, and thus the water solubility and swelling of the protein film, as well as its WVP, have decreased [136,262].

Amazing features could be achieved by the insertion of MNP or MONPs as nanofillers. Metallic nanoparticles are promising in active packaging systems for innovative technology in food preservation based principally on mass transfer interactions between food and packaging materials [268]. Cano’s research team [269] developed biodegradable starch–PVA films containing AgNPs with antimicrobial effect against bacteria (*Listeria innocua* and *Escherichia coli*) and fungi (*Aspergillus niger* and *Penicillium expansum*). They observed that the silver ions’ release was slowed down in the nonpolar medium (involving oleic acid), suggesting the usage of this film in fat-rich products.

Obtaining green MNPs opened a new era in nanotechnology. Vegetal extracts are a good choice for “green” synthesis of metallic nanoparticles, since plant extracts contain many bioactive compounds (aminoacids, proteins, enzymes, polysaccharides, organic acids, polyphenols, flavonoids, terpenoids, tannins, etc.) that can bioreduce and stabilize the nanoparticles [270]. We could choose a plant or a plant mixture, taking into account their health benefits. Plants are most abundant in nature, they are renewable, and they are ecofriendly. Plants contain a “cocktail” of active ingredients which could help to sustain and stimulate human immunity; hence it is recommended to use phytosynthesized nanoparticles in food packaging composite nanomaterials, since phytonanometals bear on their surface active compounds derived from plants, which could have beneficial effects on health due to their antioxidant and antimicrobial properties [118,120,271,272,273,274,275].

Figure 6 displays a suggestive schematic representation of “green” design starch-based food packaging bionanocomposites containing phytosynthesized MNPs.

Many research studies are needed to evaluate the nanomaterials’ cytotoxicity. Nanosized systems can migrate from package to food, resulting in serious health concerns. The migration rate increases if the nanoparticle size and polymer viscosity decrease, and if the nanoparticles’ concentration increases [276,277]. If the nanomaterials are too small, they can enter the circulatory system and cause health risks; therefore it is better to optimize the size and the concentration of the nanomaterials in order to avoid these problems. Some reports related to use of AgNPs, ZnONPs and SiO_2_NPs proved that only a small number of particles migrated from nanocomposites packaging to foods, being below the limits prescribed by the European Commission [276]. Abreu et al. [278] prepared starch film nanocomposites loaded with AgNPs and montmorillonite modified with a quaternary ammonium salt (C30B); these nanostructured films exhibited improved mechanical and gas barrier properties, and also showed good antimicrobial results (similar to higher AgNPs level). The authors found that the migration of components from these obtained starch films to food was under the legal limits.

In the last few years, ecofriendly products have attracted wide interest as safe and nontoxic food packaging materials, but further improvements are also needed to develop optimal formulations for bionanocomposites that comply with the regulations relating to the migration of nanomaterials from packaging to food [154,279].

## 7. Frontier Technologies in Green Starch-Based Films and Coatings Research

Recently, bioinspiration became a new trend in green technology. As is known, the structure and optical properties are very important features of food packaging systems, so biomimetics offer new interesting aspects in this regard, since they have regular and self-assembled layered micro-/nanostructures which confer characteristic optical properties at the interaction with light [280], and also self-healing ability.

A source of inspiration from nature is nacre (“mother-of-pearl”), a material fabricated by pearl oysters, being remarkable for its highly regular “brick-and-mortar” arrangement of the alternatively packed aragonite (CaCO_3_) plates and the biomatrix (chitin, proteins) [280,281,282]. Scientists designed nacre-mimetic artificial materials, especially in combination with clay (such as montmorillonite), exhibiting excellent mechanical properties and good thermal stability, for applications in food packaging [282] among others. Montmorillonite (MMT) is one of the most extensively utilized natural clay minerals due to its low-cost, good biocompatibility, high specific surface area, excellent TS and good thermal stability, and gas barrier properties [283,284]. MMT belongs to the smectite group, encompassing two fused siloxane tetrahedral sheets sandwiching an edge-shared octahedral sheet made up of aluminum or magnesium hydroxide [285]. 

Inspired from the amazing ability of mussels to adhere to various kinds of surfaces via marine mussel adhesive proteins (MAPs), the researchers designed biomimetic adhesive coatings from polydopamine (PDA) [281], which is a biomolecule with remarkable adhesion properties due to catechol and amine groups—it is obtained by the polymerization of dopamine (a neurotransmitter in the brain, and one of the important mediators of neuroimmune interactions) [286]. 

By combining the two biomimetic approaches, nacre and MAPs, researchers designed, by ecofriendly methods, starch-based biocomposites with improved mechanical properties (TS, Young’s modulus of the starch film; stronger interactions between fillers and starch matrix). Thus, the first study regarding composite materials of starch and MMT modified with polydopamine, was reported by Zhou and Xu, in 2015 [283]. In 2016, Zhou’s research team [284] obtained corn starch/chitosan films reinforced with montmorillonite (MMT) modified by bioinspired polydopamine, with enhanced effective stress transfer between MMT and polymer matrix. Li et al. [281] developed a starch-based nacre-mimetic nanocomposite by assembling on corn starch, MMT coated with a thin layer of polydopamine (PDA), resulting in strong interfacial adhesion between the filler and the matrix, and also improved mechanical properties. 

“Lotus effect” is another remarkable biological model for bioinspiration in food packaging applications. Starting from the studies of Barthlott and Neinhuis in 1997 [287], the researchers who discovered the unique nonwettable and self-cleaning properties of the lotus leaf, various strategies were used to develop superhydrophobic materials. Because the surfaces with few or no polar groups exhibit a very low interfacial tension [288], one idea to reduce starch hydrophilicity was to crosslink its hydroxyl groups with a hydrophobic polymer. Chen et al. [285] highlighted that there are two key points in achieving superhydrophobic surfaces: low surface energy and multiscale surface roughness. In their study, Chen’s research team developed starch-based composites containing: (i) polydimethylsiloxane—used to decrease the starch film surface energy, and to crosslink with the hydroxyl group on the starch surface, and (ii) montmorillonite (MMT)—used to integrate roughness in micro- and nanoscale [285].

## 8. Conclusions

In this paper we review new research trends that present starch, a natural sustainable biopolymer, as a primary green packaging material to be used as films or coatings on food. Starch is used here as model to connect the concept of Green Packaging (GP), along with another two “green” concepts, namely the Green Chemistry (GC) and Green technology (GT), by incorporating bioactive compounds into the starchy matrix and by using physical treatments which are environmentally friendly on the films resulted or for the obtaining of packaging. 

In relation to GC, the focus is on the characterization and incorporation of the results into the starch-based films or coatings of the biocompounds involved in supporting health: vitamins, minerals, polyphenols, essential oils, amino acids, proteins and lipids, isolated from plants or in the form of plant extracts, rich in mixes of these bioactive compounds. Some compounds, such as vitamins or minerals, are not yet extensively used. By comparison, studies regarding the use of essential oils, polyphenols, proteins and lipids, and their effect on food coated or packed into starch-based films containing them are numerous. A large number of studies regarding the incorporation of plant extracts directly into the starch packaging, with very good results on the antimicrobial, antioxidant and/or mechanical properties of the films resulted and/or the food where the film was applied as primary packaging, are published. The development of starch-based primary films or coatings incorporating compounds with biological effect on health could be the next revolution in GP. 

As a way to connect GP to GT through the starch-based packaging, green physical techniques such as UV and ionizing radiation, together with atmospheric pressure cold plasma, are presented. Generally, radiation improves the mechanical properties of films, but this action is dependent on the radiation type and dose as well as the treatment time. Atmospheric pressure cold plasma induces various modifications directly into the starch granule, thus making this treatment method a promising technology to obtain films or coatings; however, investigations are only just beginning in this research field.

Moving forward, we also propose to expand the concept of GP to another challenging field—that of Green Nanotechnology (GN)—by showing the actual status in developing green nanopackaging. The research is still in its initial stage and only a few publications on starch-based nanopackaging were found, with good results in improving the films’ mechanical properties and antimicrobial activity.

In the end, our review presents a few proposals regarding using the nature as source of inspiration for future research, such as nacre-mimetics in combination with clay, mussel adhesive proteins or superhydrophobic materials to be combined with starch with the aim of improving its characteristics and developing a new generation of superpackaging.

## Figures and Tables

**Figure 1 polymers-13-01190-f001:**
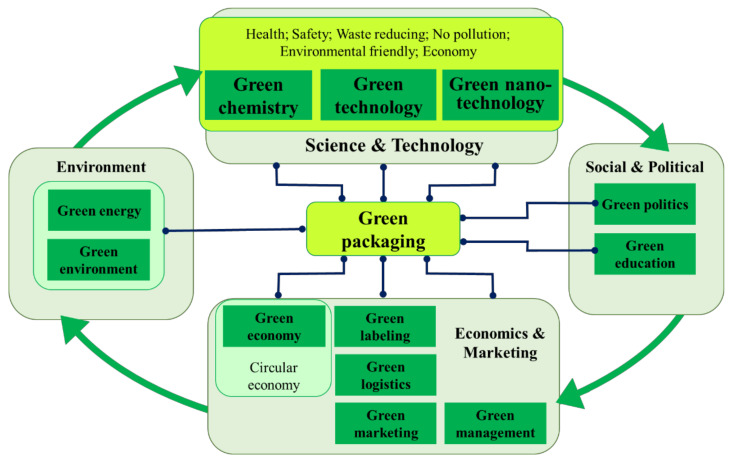
Connections between the concept of green packaging and other “green” concepts (original concept drawing based on the literature research from [10,11,12,13,14,15,16,17,18,19,20]).

**Figure 2 polymers-13-01190-f002:**
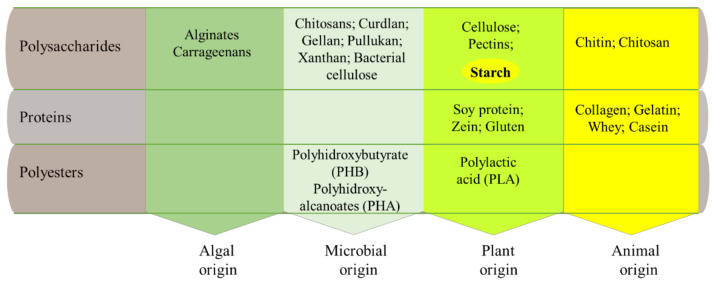
Classification of biopolymers depending on the general chemical composition and their origin (original drawing based on the literature research from [31,32,33,34,35]).

**Figure 3 polymers-13-01190-f003:**
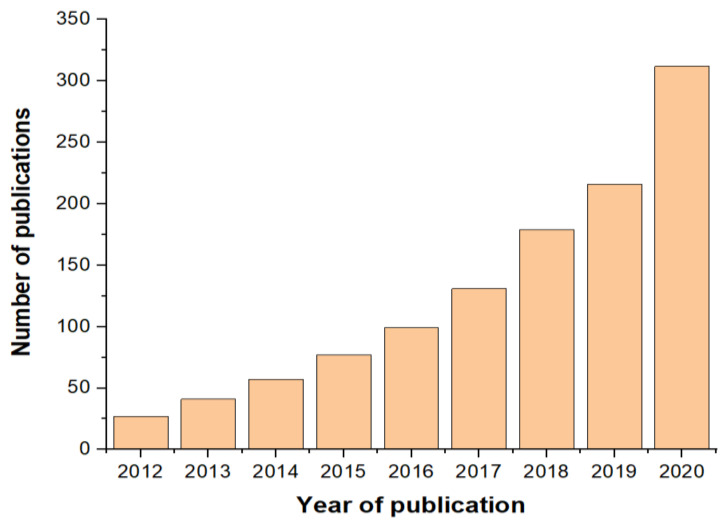
Evolution in the last decade, of publications related to “starch food packaging” (graphic generated with the results from Web of Science (WOS) by using the keywords “starch food packaging”).

**Figure 4 polymers-13-01190-f004:**
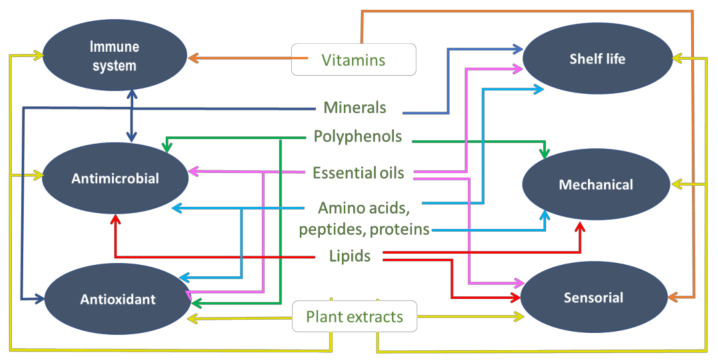
Overview on the main actions of starch-based packaging incorporating bioactive compounds and plant extracts for sustaining consumers’ health and safety (**left**) and the packaged products (**right**) (original drawing by the authors).

**Figure 5 polymers-13-01190-f005:**
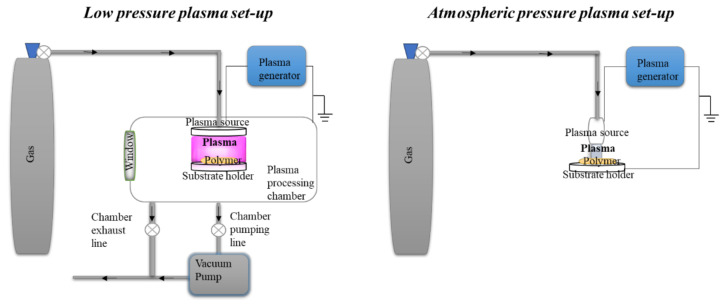
Comparison between low and atmospheric pressure plasma set-up used for polymers treatments, an original drawing based on the operation principles of plasma set-ups found in references [248,249].

**Figure 6 polymers-13-01190-f006:**
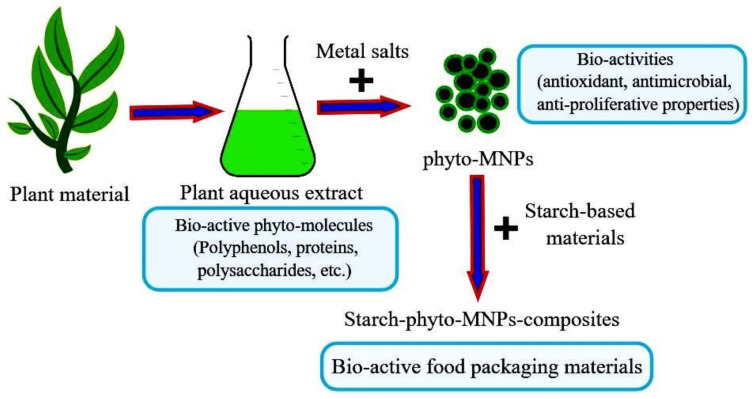
Schematic representation of “green” bottom-up approach to develop bioactive starch food packaging materials based on phytosynthesized metallic nanoparticles (phyto-MNPs). Figure was created with Chemix (https://chemix.org/, accessed on 15 November 2020).

**Table 1 polymers-13-01190-t001:** Applications of biodegradable films incorporating polyphenols, used as food packaging.

Source	Film Matrix	Polyphenols Concentration	Application	References
Tea	Pomelo peel flours	5–20%	Soybean oil preservation	[82]
Tea	Alginate	1–5%		[78]
Tea	Starch	0.06–0.6%		[83]
Green tea extract	Starch	5%	Beef	[139]
Apple peel	Chitosan	0.25–1%		[138]
Young apple polyphenols extract	Chitosan	0.25–1%		[140]
Grapefruit seed extract	Rapeseed protein–gelatin	1%	Strawberries	[141]
Grapefruit seed extract	Agar/alginate/collagen hydrogel	0.2%	Potatoes	[142]
Yerba mate extract and mango pulp	Cassava starch	Mango pulp 20% Yerba mate extract 30%	Palm oil	[143]

**Table 2 polymers-13-01190-t002:** The effect of atmospheric pressure cold plasma upon starch.

Starch Type and Form	Plasma Experimental Conditions	Main Findings after Plasma Treatment	Ref.
Tapioca starch tablets	High voltage dielectric barrier discharge (DBD), 17 kHz frequency of power supply, 40 watts, treatment time 30 min	Higher degree of crystallinity in starch for high humidity conditions, increase in the degree of crosslinking, for all humidity conditions	[250,251]
Tapioca starch slurry	Atmospheric pressure argon plasma jet, 600 MHz high frequency, 50 or 100 W power, treatment time 5 min	Cross-linking or depolymerization of starch determined by the preparation of starch slurry and the plasma input power	[246]
Corn starch	Dielectric barrier discharge plasma, 50 V voltage, 1.5 A current, 75 W power, air gas, treatment time 1, 5 or 10 min	Larger channels of the starch granules, decrease in the degree of crystallinity, oxidation of partial hydroxyl groups to carboxyl groups, and molecular degradation, the viscosity decreased	[252]
Rice, potato, tapioca and corn starch films	High voltage dielectric barrier discharge (DBD) atmospheric cold plasma, frequency of 60 Hz, voltage 80 kV, treatment time 5 min	Increase in the glass transition temperature, surface roughness and surface oxygenation, the amylose content and the starch source play an important role in determining the polymer’s interaction with cold plasma	[253]
Granular cassava starch	Atmospheric dielectric barrier discharge (DBD) plasma, argon gas, electric current of 1.0 A, power supply 4–9 kV, treatment time 0–40 min	Increase in the crosslinking, effects in morphological properties, treated starch became highly resistant to enzymatic hydrolysis leading to the increasing of resistant starch content	[254]
Maize starch powder	Dielectric Barrier Discharge (DBD) cold argon-plasma treatment at atmospheric pressure, input parameters: 1.0 A, 176 V and 50 Hz, treatment time 10 min	Increase in crystallinity, reduction of rapidly digestible starch, water absorbance index and swelling factor, reduced molecular weight, more compact in structure than its raw starch	[255]
Waxy maize starchand normal maize starch as suspension	Atmospheric pressure plasma jet, 750 W input power, 25 kHz frequency of power supply, treatment time 1, 3, 5, or 7 min	Slight breakage of the surface of the starch granules, increases in waxy maize starch and swelling volume, and decreases in gelatinization temperature and enthalpy, decreases the relative crystallinity, reduces short-range molecular order	[247]
Banana starch suspension	Corona electrical discharge (CED), current intensity of 60 A at 30 kV/cm, 40 kV/cm, and 50 kV/cm, treatment time 3 min	Surface damage of the starch granules, reduction in the total area of diffraction peak, gelatinization enthalpy, and different pasting behaviors	[256]
Potato starch slurry	Atmospheric pressure plasma jet, power supply of 750 W, 25 kHz frequency of power supply, treatment time 1, 3, 5 or 7 min	Decreased relative crystallinity and short-range molecular order, slight damage in starch granule morphology	[257]
Maltodextrin (incomplete hydrolysate of starch) powder	Argon-plasma cold dielectric barrier discharge (DBD) at atmospheric pressure, 1 ampere, 120 voltages, 50 Hz, treatment time 0, 5, 10, 15 and 20 min	Reduce the level of polymerization and molecular weight, increase dextrose equivalents (higher sweetness)	[258]
Tatary buckwheat, quinoa and sorghum dry starches	Atmospheric plasma operated at 20 kV and at a frequency of 1 kHz, treatment time 30 s	Reduced amylose content and swelling power, and higher relative crystallinity, pasting temperature and syneresis, different surface modifications depending on the starch type, dramatic decrease in viscosities, higher degree of hydrolysis	[259]
Corn starch (S-41260) powder	Atmospheric pressure air cold plasma jet, high voltage input powers: 400 W, 600 W, and 800 W, treatment time 30 min	Production of small molecular fragments and hydrophilic functional groups, reduction in viscosity and an increase in solubility and starch paste clarity	[260]
Banana starch suspension	Atmospheric cold plasma dielectric barrier discharge, 30–50 V, treatment time 3 min	The amylose content increase as the treatment intensity increased, decomposition of the outer layers of banana starch granules depending on the treatment intensity, partial decomposition and etching of DBD treatment	[261]
Cassava starch colloid	Direct current (DC) pulsed plasma, 16 kV voltage, 3 µs pulse width, 20–30 kHz applied pulsed frequency, treatment time 0–300 min	Increased carbonyl and carboxyl groups of oxidized cassava starch, increase in hydroxyl radicals with increasing treatment time and pulsed frequency, reduction in amylose content	[197]

## Data Availability

Not applicable.

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
