# Peer review of "Green Design of Novel Starch-Based Packaging Materials Sustaining Human and Environmental Health"

_polymers, 2021, doi:10.3390/polym13081190_

Round 1
Reviewer 1 Report
The novelty of the work is questionable because I have read several papers on similar topic. There are many articles already published on the similar topic and this one follows the same structure and content as many of the previously published ones, without adding much of significance to the current knowledge. The work brings nothing new to learning. A review paper should give a critical overview. So, I will not recommend the acceptance for this manuscript in the current form. Authors must include some more good quality figures related to the content of the manuscript.
Author Response
We appreciate the Reviewer comment. Please find below a detailed answer to the comment.
There are many reviews related with the obtaining of biopolymers-based films (starch-based films, too) and other related with the incorporation of different biocompounds into these films (starch-based or not). Some of them are cited in our review. The novelty and significance of our approach lays in the fact that this is the first critical overview integrating the concept of green packaging with three other “green” approaches, namely green chemistry, green technology and green nanotechnology; more, in this review we present the state-of-the-art regarding starch-based films used in food packaging that incorporates green chemical bioactive compounds that normally help boosting our health, are environmentally friendly and are obtained only by green chemical and/or physical synthesis/processing methods. There is no such original approach regarding starch-based packaging published in the scientific literature that include so many aspects. We pointed out this original aspect in the abstract as well as in the chapter ”1. The “green” context” (lines 103-123 in the first version of the manuscript) and then we overviewed the existent literature in the next chapters. We justified our choice on the selected green chemical and physical treatments.
Besides reviewing different solutions regarding the incorporation of bioactive ingredients into the starch matrix, original issues and novel ideas are included, as inducing various modifications directly into the starch granule by using atmospheric pressure cold plasma (in order to obtain films or coatings) Ë— investigations are only at the beginning in this research field. To the best of our knowledge such a review paper is not yet found in literature, no other review focus on the non-thermal physical starch treatment methods, too.
In addition, at the end of the manuscript, future perspectives are given regarding the research topic: Bioinspiration ideas Ë— the biomimetic construction of future green starch-based packaging; use of phyto-synthesized MNPs for “green” starch-based food packaging bionanocomposites.
Nevertheless, your critical review stimulates us to improve our manuscript in order to show the specificity and the novelty of our work. Therefore, the major improvements are given below:
Improvement 1:
We shortened the title of the manuscript, from:
“Green design of novel starch-based packaging as films and coatings sustaining human and environmental health”
To:
“Green design of novel starch-based packaging materials sustaining human and environmental health“
Improvement 2:
We improved the abstract, please find it below:
A critical overview of current approaches to the development of starch-containing packaging, integrating the principles of green chemistry (GC), green technology (GT) and green nanotechnology (GN) with those of green packaging (GP) to produce materials important for both, us and the planet, is given. First, as a relationship between GP and GC, the benefits of natural bioactive compounds are analyzed and the state-of-the-art is updated in terms of starch packaging incorporating green chemicals that normally help us to maintain healthy, are environmentally friendly and are obtained via GC. Newer approaches are identified, as the incorporation of vitamins or minerals into films and coatings. Second, the relationship between GP and GT is assessed by analyzing the influence on starch films of green physical treatments such as UV, electron beam or gamma irradiation, and plasma; emerging research areas are proposed, as the use of cold atmospheric plasma for films production. Thirdly, approaches on how GN can be used successfully to improve the mechanical properties and bioactivity of packaging are summarized; current trends are identified as a green synthesis of bionanocomposites containing phyto-synthesized metal nanoparticles. Last but not least, bio-inspiration ideas for the design of the future green packaging containing starch are presented.
Improvement 3:
We re-written the keywords:
green packaging; green chemistry; green technology; starch; bioactive; health; UV; cold plasma; nanopackaging; bioinspiration
Improvement 4:
We improved Figures 1, 2 and 3. In the case of Figure 1, we evidenced our focus on the inter-relation between green packaging and three other ”green” concepts from all those presented in the Figure 1. In the case of Figure 3, we improved the resolution. For a better resolution, we highlighted only the evolution of publications in the last decade.
We included a new figure, noted as Figure 4 in the revised document.
We specified that ALL figures in this review are original and created by us.
In conclusion, please reconsider your score regarding our manuscript.

Reviewer 2 Report
The traditional food packaging materials are mainly manufactured by petroleum-derived polymers, which cause tremendous damage to the environment, water supplies and the entire ecosystem. At present, the food packaging products derived from polysaccharides (cellulose and starch) and bio-based plastics have drawn consideration attention. Additionally, the present food packaging materials also have many functional requirements (particularly human and environmental health benefits). Considering the above-mentioned backgrounds, authors review the feasibility and advantages of starch-based packaging. The review focuses on the incorporation of the bioactive compounds into the starch matrix, the green chemical and physical treatments of the starch-based packaging, the application of green nanotechnology, and the biomimetic construction of future green starch-based packaging.
Overall, the manuscript is well organized, the contents are comprehensive, and the important viewpoints and novel ideas are clearly presented. Also the English expression is fluent and accurate. The manuscript is a good review. The manuscript can be accepted for publishing after minor revision.
Please note the following minor issues:
Regarding titles: I can not find the sixth titles.
5. Application of green physical treatments on starch and starch-based films
7. Nanotechnology in starch food packaging
Abstract:
Lines 21-23: “The health benefits of some important natural bioactive compounds --- together with plant extracts --- is revealed --- ”: “is revealed” should be replaced by “are revealed”.
1. The “green” context
Line 43: “Green packaging (GP) has his place inside this paradigm”: “his” should be replaced by “its”.
Lines 50-51: “GP has as goal the safe packaging for a better life on Earth and, generally, refers to packaging design”: I suggest authors to improve this sentence.
2. Starch-based films and coatings
“Starch can be found in different resources such as wheat, maize, potato, bean, rice and other”: “other” should be replaced by “others”.
3. Plant-derived bioactive compounds promoting the human health
Lines 213-215: “leading to multiple health-promoting effects like improved immunity, reduced inflammation, blood-brain barrier integrity, proper functioning of gut [63]”: Between “blood-brain barrier integrity”and “proper functioning of gut”, the word “and” should be added.
The content regarding the human health benefits of various Plant-derived bioactive compounds is too long, which weakens the subject of packaging.
4. Green chemical treatments of the starch-based films and coatings by incorporating bioactive compounds
This title contains “by incorporating bioactive compounds”. However, sub-titles are as follows: 4.1. Vitamins incorporation into starchy films and coatings, 4.2. Polyphenols, 4.3. Essential oils, 4.4. Minerals, 4.5. Amino acids, peptides, proteins and enzymes in starch-based food packaging, 4.6. Lipids and lipid-based nanostructures in starch packaging systems, 4.7. Vegetal extracts in starch food packaging.
Minerals and some proteins are bioactive compounds?
Lines 558-559: “whereas WVP can be decrease by adding hydrophobic materials such as oleic acid or beeswax”: “decrease” should be replaced by “decreased”.
Lines 561-563: “On the other hand, the inclusion of nanofillers --- improves mechanical strength, thermal stability, and the water and oxygen barrier properties [154].”: “and the water and oxygen” should be replaced by “and water and oxygen”.
Line 564: “Starch–Proteins” should be replaced by “Starch–proteins”.
5. Application of green physical treatments on starch and starch-based films
Line 734: “in the last two decades’”: The superscript [’] should be deleted.
7. Nanotechnology in starch food packaging
In the section, nanoparticles are discussed. However, in the 4.4 Minerals section, some nanoparticles are also mentioned.
8. Open issues for starch-based films and coatings research
“Open issues for starch-based films and coatings research” is a proper title? The contents seem to be frontier technologies.
Author Response
Thank you very much for your appreciations and for your suggestions! Please find bellow detailed answers to each of the comments:
Comment 1:
Regarding titles: I can not find the sixth titles.
- Application of green physical treatments on starch and starch-based films
- Nanotechnology in starch food packaging
Answer 1:
Thanks for this observation! Indeed, the sixth title is missing and we corrected the titles numbers. We renumbered the sections as follow:
- 6. Nanotechnology in starch food packaging
- 7. Open issues for starch-based films and coatings research
- 8. Conclusions
Comment 2:
Abstract:
Lines 21-23: “The health benefits of some important natural bioactive compounds --- together with plant extracts --- is revealed --- ”: “is revealed” should be replaced by “are revealed”.
Answer 2:
Indeed, the text needs corrections. In the Abstract, we replaced “is revealed” by “are revealed”.
Comment 3:
- The “green” context
Line 43: “Green packaging (GP) has his place inside this paradigm”: “his” should be replaced by “its”.
Answer 3:
Indeed, this correction is necessary. Changes in manuscript: “Green packaging (GP) has its place inside this paradigm”
Comment 4:
Lines 50-51: “GP has as goal the safe packaging for a better life on Earth and, generally, refers to packaging design”: I suggest authors to improve this sentence.
Answer 4:
Indeed, the phrase could have been better formulated. As suggested, we improved this sentence as follows:
“GP refers to a safe packaging design with low environmental impact”
Comment 5:
- Starch-based films and coatings
“Starch can be found in different resources such as wheat, maize, potato, bean, rice and other”: “other” should be replaced by “others”.
Answer 5:
Indeed, the text needs corrections. We replaced: “other” by “others”.
Comment 6:
- Plant-derived bioactive compounds promoting the human health
Lines 213-215: “leading to multiple health-promoting effects like improved immunity, reduced inflammation, blood-brain barrier integrity, proper functioning of gut [63]”: Between “blood-brain barrier integrity”and “proper functioning of gut”, the word “and” should be added.
Answer 6:
Indeed, the text needs corrections. As suggested, we inserted “and” between “blood-brain barrier integrity” and “proper functioning of gut”, in the lines 213-215.
Comment 7:
The content regarding the human health benefits of various Plant-derived bioactive compounds is too long, which weakens the subject of packaging.
Answer 7:
Indeed, the chapter is long because we have tried to point out the main interesting and beneficial effects of the plant-derived bioactive compounds on human health. We took into consideration your comment and we shortened this chapter by keeping the information we consider important to sustain the choice of specific bioactive compounds with benefits for health as appropriate for incorporating into the starch-based packaging.
Comment 8:
- Green chemical treatments of the starch-based films and coatings by incorporating bioactive compounds
This title contains “by incorporating bioactive compounds”. However, sub-titles are as follows: 4.1. Vitamins incorporation into starchy films and coatings, 4.2. Polyphenols, 4.3. Essential oils, 4.4. Minerals, 4.5. Amino acids, peptides, proteins and enzymes in starch-based food packaging, 4.6. Lipids and lipid-based nanostructures in starch packaging systems, 4.7. Vegetal extracts in starch food packaging.
Minerals and some proteins are bioactive compounds?
Answer 8:
Yes, minerals and some proteins are bioactive compounds. Bioactive compounds are agents that can present therapeutic potential, having a positive influence on human health. They are being intensively studied to evaluate their effects on health, and bioactive compounds appear to have beneficial physiological, behavioral, and immunological effects. Some examples of bioactive compounds are: carotenoids, flavonoids, carnitine, choline, coenzyme Q, phytosterols, phytoestrogens, glucosinolates, and polyphenols. Since vitamins and minerals elicit pharmacological effects, they can be categorized as bioactive compounds as well [https://www.sciencedirect.com/topics/agricultural-and-biological-sciences/bioactive-compound].
Bioactive peptides are defined as peptide sequences within a protein that exert a beneficial effect on body functions and/or positively impact human health, beyond its known nutritional value. These peptides can regulate important bodily functions through their myriad activities, including antihypertensive, antimicrobial, antithrombotic, immunomodulatory, opioid, antioxidant, and mineral binding functions [https://www.ncbi.nlm.nih.gov/pmc/articles/PMC6265732/; doi:10.3390/foods9070846; doi: 10.1016/s1043-4526(03)47004-6.].
We mention for each particular mineral its therapeutic potential when embedded into a packaging product such as antimicrobial activity or antioxidant and antibacterial activities.
Indeed, some proteins that first improve the physical properties of films are presented in the "bioactive compounds" section, but we have also highlighted their therapeutic potential.
Comment 9:
Lines 558-559: “whereas WVP can be decrease by adding hydrophobic materials such as oleic acid or beeswax”: “decrease” should be replaced by “decreased”.
Answer 9:
Thanks for this observation! We replaced “decrease” by “decreased”, in Lines 558-559.
Comment 10:
Lines 561-563: “On the other hand, the inclusion of nanofillers --- improves mechanical strength, thermal stability, and the water and oxygen barrier properties [154].”: “and the water and oxygen” should be replaced by “and water and oxygen”.
Answer 10:
Indeed, the text needs corrections. We replaced “and the water and oxygen” by “and water and oxygen”, in Lines 561-563.
Comment 11:
Line 564: “Starch–Proteins” should be replaced by “Starch–proteins”.
Answer 11:
Indeed, the text needs this correction. We replaced “Starch–Proteins” by “Starch–proteins”.
Comment 12:
- Application of green physical treatments on starch and starch-based films
Line 734: “in the last two decades’”: The superscript [’] should be deleted.
Answer 12:
Indeed, the text needs corrections. We deleted the superscript [’] in the Line 734: “in the last two decades’”.
Comment 13:
- Nanotechnology in starch food packaging
In the section, nanoparticles are discussed. However, in the 4.4 Minerals section, some nanoparticles are also mentioned.
Answer 13:
Thank you very much for this observation! In the section 4.4 Minerals, we deleted the sentence:
“Silver (Ag) particles have shown antimicrobial and fungicidal activity.”
and we moved the paragraph:
“A group of researchers incorporated ZnO nanoparticles in different films made from agar, carrageenan and carboxymethyl cellulose. They observed that nanoparticles were uniformly distributed into the matrix and the colors of the films were modified by ZnO nanoparticles. The incorporation of ZnO nanoparticles into the films led to increased hydrophobicity, UV barrier, Eb value, moisture content and thermal stability of the films, whilst the water vapor barrier and TS were lower. The films with ZnO nanoparticles showed antimicrobial effects against L. monocytogenes and E.coli [150].”
to the section 6 (Nanotechnology in starch food packaging).
In the section 4.4 Minerals, we inserted the sentence:
“Moreover, the antimicrobial properties of silver and zinc oxide in nanoparticulate forms are exploited in development of packaging materials as described further in the section 6 (Nanotechnology in starch food packaging).”
Comment 14:
- Open issues for starch-based films and coatings research
“Open issues for starch-based films and coatings research” is a proper title? The contents seem to be frontier technologies.
Answer 14:
Indeed, the title needs adjustments, therefore, we slightly modified it as follows:
- Frontier technologies in starch-based films and coatings research

Reviewer 3 Report
Comments to the authors:
The manuscript is entitled “Green design of novel starch-based packaging as films and coatings sustaining human and environmental health". Monica Minorescu et al. had shown in their manuscript the principles of green chemistry, green technology and green nanotechnology, green packaging can produce materials with an important role in supporting our health and the environment. The approaches to design starch-based green food packaging that incorporates green chemical bioactive compounds that normally help boosting our health. The health benefits of some important natural bioactive compounds together with plant extracts is revealed and the results on the incorporation of these bioactive compounds into the starchy matrix. Moreover, this review includes the influence on the starch-based films of green physical treatments such as UV radiation and atmospheric cold plasma. Green nanotechnology can be successfully used to improve the mechanical properties and the bioactivity of starch-based films. This manuscript can be accepted for publication in Polymers Journal, however, it needs some simple adjustments. I recommend Accept after minor revision. After a critical evaluation of the manuscript, some comments are done as follows:
Abstract:
I suggest that you remove from the abstract the words that are in the title paper´s title.
Introduction:
Page 2, line 53: I suggest you remove the “too” word, it is not necessary.
Page 2, line 71, 64 and 44: Standardize “figure 1” or “Figure 1.” It is not in accordance with manuscript.
Page 2, Figure 1: Why is Circular economy in a different color? To highlight something? If so, add it to the text.
Page 3, line 111: Insert only the acronyms GC and GT, as they were previously mentioned in line 44.
Starch-based films and coatings:
In this topic, you could briefly mention which methods are most used for the formation of films and coatings.
As it is a review that the mentioned biopolymer is the starch, wouldn't it be important to show its structure? I suggest this starch's review article. DOI: 10.3126/ije.v4i4.14108.
Page 4, line 137. In phrase: “Another important characteristic is its biocompatibility with many other biopolymers”. Is this term biocompatibility correct?
Page 4, line 152. In phrase: “heat treating aqueous mixture of 100 parts starch with 25-100 parts glacial acetic acid at 100°C”. Did the authors mean % w/w? Please insert it as the authors of the manuscript [46].
Page 5, line 185: The word "boom" is not appropriate. Please readjust this word.
Page 5, Figure 3: The numbers in this figure are hard to visualize it. Please improve the figure resolution.
Vitamins incorporation into starchy films and coatings:
Page 10, line 460: In phrase: “Fakhouri et al. described the manufacturing process of the film with ascorbic acid from cranberry powder and evaluated its characteristics”. I think it would be important to mention which characteristics have been evaluated and improved with vitamins' incorporation.
Page 14, line 644 and 662: I do not see the correct use of ‘‘and’’, not only following the lines cited but also in some parts of the text. Please check this.
Author Response
We are thankful to the Reviewer very pertinent observations. Please find bellow detailed answers to each of the comments:
Comment 1:
Abstract:
I suggest that you remove from the abstract the words that are in the title paper´s title.
Answer 1:
According to your suggestion we modified the title and the abstract. Thus, the title
“Green design of novel starch-based packaging as films and coatings sustaining human and environmental health”
became:
“Green design of novel starch-based packaging materials sustaining human and environmental health“
Where possible, we removed from the abstract words that are in the paper´s title, too, please analyse it again.
Abstract: A critical overview of current approaches to the development of starch-containing packaging, integrating the principles of green chemistry (GC), green technology (GT) and green nanotechnology (GN) with those of green packaging (GP) to produce materials important for both, us and the planet, is given. First, as a relationship between GP and GC, the benefits of natural bioactive compounds are analyzed and the state-of-the-art is updated in terms of starch packaging incorporating green chemicals that normally help us to maintain healthy, are environmentally friendly and are obtained via GC. Newer approaches are identified, as the incorporation of vitamins or minerals into films and coatings. Second, the relationship between GP and GT is assessed by analyzing the influence on starch films of green physical treatments such as UV, electron beam or gamma irradiation, and plasma; emerging research areas are proposed, as the use of cold atmospheric plasma for films production. Thirdly, approaches on how GN can be used successfully to improve the mechanical properties and bioactivity of packaging are summarized; current trends are identified as a green synthesis of bionanocomposites containing phyto-synthesized metal nanoparticles. Last but not least, bio-inspiration ideas for the design of the future green packaging containing starch are presented.
Comment 2:
Introduction:
Page 2, line 53: I suggest you remove the “too” word, it is not necessary.
Answer 2:
Indeed, this correction is necessary. We eliminated the word “too”.
Comment 3:
Page 2, line 71, 64 and 44: Standardize “figure 1” or “Figure 1.” It is not in accordance with manuscript.
Answer 3:
Thank you for this observation! We adopted the notation: “Figure 1”.
Comment 4:
Page 2, Figure 1: Why is Circular economy in a different color? To highlight something? If so, add it to the text.
Answer 4:
Thank you for this observation! We included a sentence explaining why the strategy of circular economy, which is part of the green economy:
(according to the principle strategy of circular economy, a practical green economic solution)
We improved the Figure 1 by including the circular economy into the general concept of green economy, to correspond better to the approach.
Comment 5:
Page 3, line 111: Insert only the acronyms GC and GT, as they were previously mentioned in line 44.
Answer 5:
Indeed, the text needs correction. According to your suggestion we corrected the text. We inserted only the acronyms GC and GT, at Page 3, line 111.
Comment 6:
Starch-based films and coatings:
In this topic, you could briefly mention which methods are most used for the formation of films and coatings.
Answer 6:
Indeed, there is necessary to mention which methods are the most used for the formation of films and coatings. The classical methods for the formulations of films and coatings are described at the beginning of chapter 4. Green chemical treatments of the starch-based films and coatings by incorporating bioactive compounds. We considered more adequate to include them there because the classical methods are used for the incorporation of the bioactive compounds.
We included a sentence in chapter 2. Starch-based films and coatings, to explain where the methods for the formulation of films are described:
The most used methods for producing starch-based films are extrusion and casting; they are described in Section 4. Green chemical treatments of the starch-based films and coatings by incorporating bioactive compounds.
Comment 7:
As it is a review that the mentioned biopolymer is the starch, wouldn't it be important to show its structure? I suggest this starch's review article. DOI: 10.3126/ije.v4i4.14108.
Answer 7:
We consider that the structure is quite known and that is difficult to provide an original drawing.
We cited the article indicated by you, thank you very much for the suggestion!
Comment 8:
Page 4, line 137. In phrase: “Another important characteristic is its biocompatibility with many other biopolymers”. Is this term biocompatibility correct?
Answer 8:
Indeed, the text needs corrections and the use of the term “biocompatibility” here gives rise to ambiguities. We made changes in text. At Page 4, line 137, we eliminated the “bio” as follows:
“Another important characteristic is its compatibility with many other biopolymers, as those indicated in Figure 2.”
Comment 9:
Page 4, line 152. In phrase: “heat treating aqueous mixture of 100 parts starch with 25-100 parts glacial acetic acid at 100°C”. Did the authors mean % w/w? Please insert it as the authors of the manuscript [46].
Answer 9:
Indeed, the text needs corrections. We removed the phrase, to eliminate the misunderstandings:
Starch acetylation can be accomplished by heat treating aqueous mixture of 100 parts starch with 25-100 parts glacial acetic acid at 100°C. The content of acetate groups ranges from 3-6% and depends on the amount of acid used and the duration of the treatment;
Comment 10:
Page 5, line 185: The word "boom" is not appropriate. Please readjust this word.
Answer 10:
Thank you for this suggestion! At Page 5, line 185, the sentence:
“There has been a publishing boom in the last five years, whereas recently there has been a growing interest in using starch as “green” natural resource for packaging, as well as "green" methods for materials synthesis and processing.”
was changed into:
“There have been an increasing number of publications in the last years, whereas recently there has been a growing interest in using starch as “green” natural resource for packaging, as well as "green" methods for materials synthesis and processing.”
Comment 11:
Page 5, Figure 3: The numbers in this figure are hard to visualize it. Please improve the figure resolution.
Answer 11:
Indeed, the figure resolution need to be improved. For a better resolution, we highlighted only the evolution of publications in the last decade. We improved the figure resolution.
Comment 12:
Vitamins incorporation into starchy films and coatings:
Page 10, line 460: In phrase: “Fakhouri et al. described the manufacturing process of the film with ascorbic acid from cranberry powder and evaluated its characteristics”. I think it would be important to mention which characteristics have been evaluated and improved with vitamins' incorporation.
Answer 12:
Thank you for this observation! As suggested, the sentence:
“Fakhouri et al. described the manufacturing process of the film with ascorbic acid from cranberry powder and evaluated its characteristics [131].”
was changed as follows:
“Fakhouri et al. described the manufacturing process of the starch films incorporating cranberry powder and evaluated their thermal, microstructural, mechanical, sensorial characteristics and ascorbic acid content. The ascorbic acid contained in cranberry powder improves the sensorial properties of the films, making them more attractive to tasters. The film acts as a protection for ascorbic acid, preventing its degradation by light, heat or oxidation [131].”
Comment 13:
Page 14, line 644 and 662: I do not see the correct use of ‘‘and’’, not only following the lines cited but also in some parts of the text. Please check this.
Answer 13:
Indeed, the text needs corrections. We reformulated some sentences in this regard (at Page 14, Lines 626, 644 and 662).

Round 2
Reviewer 1 Report
Accept.